# A network-based approach to identify deregulated pathways and drug effects in metabolic syndrome

Karla Misselbeck[1,2], Silvia Parolo [1]*, Francesca Lorenzini[3], Valeria Savoca[3], Lorena Leonardelli[1], Pranami Bora[1], Melissa J. Morine[1], Maria Caterina Mione [3], Enrico Domenici [1,3]* & Corrado Priami [1,4]*

Metabolic syndrome is a pathological condition characterized by obesity, hyperglycemia, hypertension, elevated levels of triglycerides and low levels of high-density lipoprotein cholesterol that increase cardiovascular disease risk and type 2 diabetes. Although numerous predisposing genetic risk factors have been identified, the biological mechanisms underlying this complex phenotype are not fully elucidated. Here we introduce a systems biology approach based on network analysis to investigate deregulated biological processes and subsequently identify drug repurposing candidates. A proximity score describing the interaction between drugs and pathways is defined by combining topological and functional similarities. The results of this computational framework highlight a prominent role of the immune system in metabolic syndrome and suggest a potential use of the BTK inhibitor ibrutinib as a novel pharmacological treatment. An experimental validation using a high fat diet-induced obesity model in zebrafish larvae shows the effectiveness of ibrutinib in lowering the inflammatory load due to macrophage accumulation.

[1] Fondazione The Microsoft Research University of Trento, Centre for Computational and Systems Biology (COSBI), Rovereto, Italy. [2] Department of Mathematics, University of Trento, Trento, Italy. [3] Department of Cellular, Computational and Integrative Biology (CIBIO), University of Trento, Trento, Italy. [4] Department of Computer Science, University of Pisa, Pisa, Italy. *email: parolo@cosbi.eu; enrico.domenici@unitn.it; priami@cosbi.eu

Metabolic Syndrome (MetSyn) is a highly prevalent pathological condition defined by a clustering of comorbidities that increases the risk of cardiovascular diseases and type 2 diabetes mellitus. The risk factors commonly associated with MetSyn are abdominal obesity, hyperglycemia, hypertension, elevated levels of triglycerides and low levels of high-density lipoprotein (HDL) cholesterol. According to the criteria proposed by the main organizations involved in the study of MetSyn, this clinical condition can be diagnosed when three of these five metabolic abnormalities are present simultaneously[1]. Additional components such as chronic pro-inflammatory and pro-thrombotic states have been repeatedly implicated in Met-Syn[2], highlighting the multifactorial nature of the disorder. While lifestyle changes are highly effective in the early phase of MetSyn, pharmacological treatments are frequently required in more advanced stages[3]. Currently, the pharmacological interventions are mostly directed towards the single MetSyn components separately, raising the problem of polypharmacy[3]. Moreover, although an altered function of adipocytes is recognized as a pivotal driver of the observed metabolic dysregulation[4], most of the drugs approved for obesity act on the central nervous system while the perturbed pathways in the adipose tissue remain less explored[5–7].

The increasing prevalence of MetSyn worldwide and the limited understanding of the pathophysiological mechanisms of MetSyn give rise to the need to study the underlying biological pathways and to develop more efficacious treatment strategies.

An effective strategy to reduce time and cost of drug development is drug repurposing (or repositioning), which identifies new therapeutic applications of already approved drugs. For example, galantamine, a drug approved for Alzheimer's disease, was recently suggested as a candidate for MetSyn therapy[8]. The growing availability of high-throughput data allows researchers to establish new computational approaches to systematically investigate drug repurposing candidates[9]. For example, signature-based methods exploiting the Connectivity Map (CMap)[10] and the Library of Integrated Cellular Signatures (LINCS) data[11] allow to identify promising candidates by comparing the transcriptomic profiles of drugs and diseases[12]. Usually, algorithms based on this approach do not consider the potential interactions among the molecular elements of the expression profiles.

Network-based analysis is a method to study in silico the complexity of biological systems and to evaluate the interactions among the different players involved, while serving as a powerful tool to link pharmacological and disease data[13]. Recent systems biology approaches based on network analysis investigated new indications for existing drugs[14–16], predicted new potential anticancer treatments[17] and identified new promising targets[18,19].

Here we propose a systems biology approach based on network integration of genomic data, text mining results, drug expression profiles and drug target information to identify the disease molecular mechanisms and to explore possible novel therapeutic strategies. Potential new therapeutic applications of already approved drugs are identified using a proximity score, which integrates a network-based distance and a functional similarity measurement.

By applying this computational framework to MetSyn, we identify the BTK inhibitor ibrutinib as a candidate drug for lowering the chronic inflammatory condition associated with obesity. Moreover, we show the effectiveness of ibrutinib treatment in lowering obesity-associated inflammation in zebrafish larvae by reducing macrophage accumulation, confirming the repurposing potential of ibrutinib in the context of obesity.

## Results

**Computational framework overview.** To obtain a systems pharmacology view of MetSyn, we devised a network-based approach that identifies functional disease modules and connects them with drug targets and drug-perturbed genes. The analytical workflow consists of three interconnected parts as shown in Fig. 1. The list of trait-associated genes was established starting

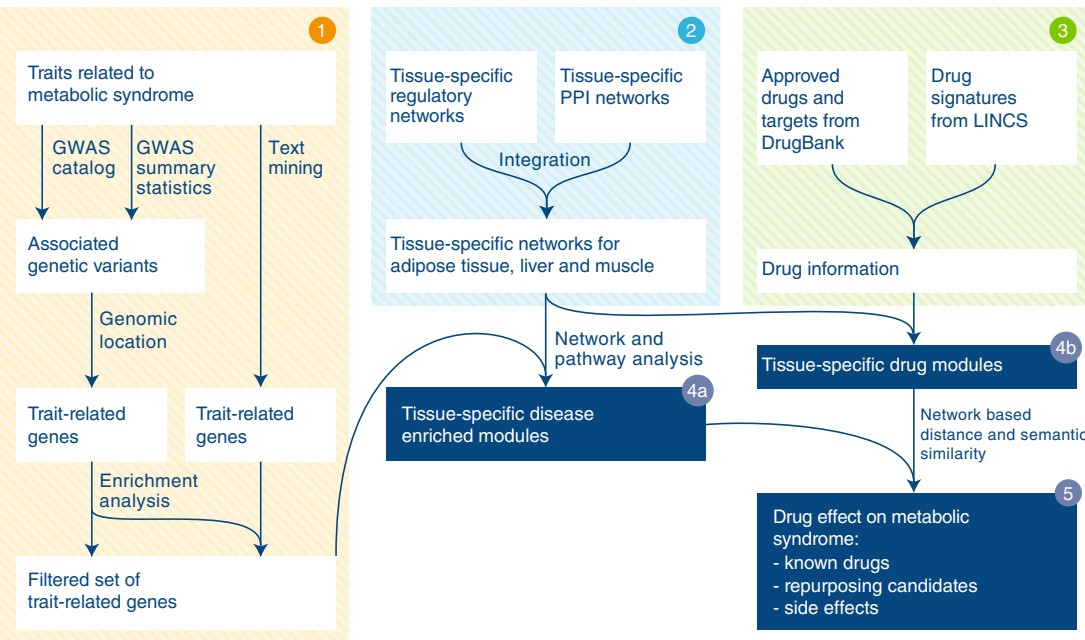

**Fig. 1** Schematic illustration of the computational framework. Step 1 MetSyn-related genes are identified by combining GWAS results and literature findings followed by a filtering step based on gene-set enrichment analysis. Step 2 Tissue-specific networks are constructed by integrating transcriptional regulatory networks from[23] and PPI networks from HIPPIE db[25]. Step 3 Drug information is retrieved from DrugBank[29] and LINCS database[11]. Step 4a and b Tissue-specific MetSyn and drug modules are established using network analysis. Step 5 To measure drug effects, a proximity score between drug and MetSyn modules is computed on the basis of network distance and semantic similarity

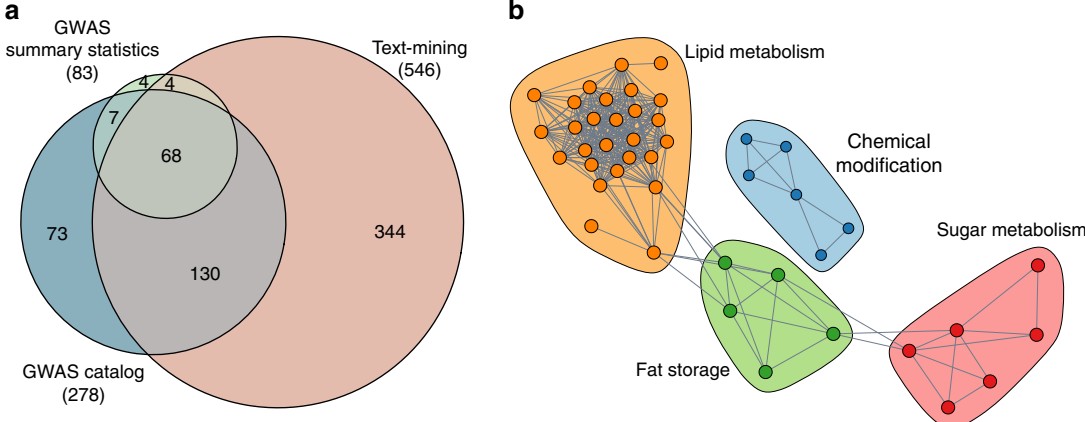

**Fig. 2** Identification of MetSyn-related genes. **a** Venn diagram showing the overlap among MetSyn genes identified using GWAS catalog, GWAS summary statistics and text mining. See also Supplementary Fig. 2 and Supplementary Data 3. **b** Pathway enrichment map showing shared gene content among the pathways enriched in MetSyn genes. Each node corresponds to a pathway and edges between pathways indicate the presence of shared genes. Colors identify the membership to communities as detected by random walk clustering algorithm. See also Supplementary Data 2

from the results of published genome-wide association studies (GWAS)[20] and from additional literature search related to metabolic syndrome (Fig. 1, step 1). Mapping the identified genes to existing biological networks (Fig. 1, step 2) allowed us to identify tissue-specific modules (hereafter called MetSyn modules), for which pathway enrichment analysis provided insight into the associated biological processes (Fig. 1, step 4a). The impact of existing drugs on MetSyn was studied by mapping drug targets and drug modulated genes on the networks in order to build drug modules (Fig. 1, steps 3 and 4b). Both drugs approved for MetSyn-related conditions and for other diseases were included in the study. This allowed us to gain insight into existing treatments and, at the same time, identify candidates for drug repositioning. To investigate the relationship between the drug modules and the MetSyn modules, we defined a proximity score that combines network-based distance with semantic similarity (Fig. 1, step 5). A drug obtains a high score if its module is close to the MetSyn module and if the genes in the drug module and the genes in the MetSyn module have a similar biological function. A comparison of our approach with previously published network-based methods for drug repurposing can be found in Supplementary Note 1.

**Identification of genes associated with MetSyn.** A widely used approach to identify genetic variants associated with common traits and diseases is the use of GWAS that over the past ten years have been applied to hundreds of phenotypes. The increasing availability of GWAS summary data permits the development of methodologies aiming at understanding the biology of phenotypes of interest starting from association results[21]. Thus, we devised a multi-step procedure to identify genes associated with MetSyn starting from the results of GWAS of relevant traits. Since the genetic variants identified by GWAS do not directly yield specific gene targets or molecular mechanisms, our workflow includes a pathway enrichment step to identify the altered biological functions. Moreover, given the incompleteness of the currently available GWAS data in explaining the heritability of traits, an external source of information (i.e. literature-derived knowledge) was included in the study. Working along this line, to generate a list of MetSyn-related genes, we combined 3 different data sources. First, we retrieved SNPs related to metabolic syndrome from the GWAS catalog (Supplementary Data 1). Despite being located mainly in introns (Supplementary Fig. 1a), the identified susceptibility variants showed a regulatory potential

since they are enriched in SNPs located in genomic regions of epigenetic chromatin marks when compared with non-selected genome-wide common SNPs (Supplementary Fig. 1b–d). We retrieved the genes located in the genomic region of the tagging SNP to assign the association signal to a gene. Second, we included the genes derived from summary statistics of 15 GWAS focused on MetSyn-related traits (Supplementary Table 1). Finally, the GWAS genes were combined with a set of genes derived from a text mining analysis performed on PubMed abstracts searching MetSyn related terms co-occurring with gene names (see Materials and Methods).

Given the heterogeneity of the data sources (GWAS catalog: top-scoring trait-associated SNPs and related genes, GWAS summary statistics: gene-level scores and text mining: genes co-occurring with MetSyn-terms) we devised a customized approach to combine and filter them. We performed a gene-set enrichment analysis of the GWAS genes selecting gene ontology biological processes and pathway databases available in EnrichR[22] to obtain pathway-level biological knowledge and selected the genes belonging to at least one significant pathway (Supplementary Data 2). In total, we were able to identify 630 genes associated with MetSyn (Supplementary Data 3, Fig. 2a, and Supplementary Fig. 2).

With this approach, we identified pathway categories such as sugar metabolism, lipid metabolism, and fat storage as significantly enriched (Fig. 2b). This annotation supports the relevance of the selected genes, as they match the pathophysiological components of MetSyn, including hyperglycemia and dyslipidemia.

**Tissue-specific disease modules.** According to the pathological phenotypes associated with MetSyn, adipose, liver and skeletal muscle were selected as trait-relevant tissues and the corresponding tissue-specific background networks were generated by combining regulatory networks and PPI networks as described in Materials and Methods (Fig. 3a). The tissue-specific regulatory networks were directly obtained from regulatory circuits[23], a resource that provides transcription factor–gene interactions inferred from the FANTOM5 data[24]. On the other hand, the tissue-specific PPI networks were created from HIPPIE interactions[25] by restricting to proteins expressed in the relevant tissue based on GTEx data[26].

The resulting network for adipose tissue contains 886 nodes and 9152 edges, the network for liver tissue 1544 nodes and 15846 edges, and the network for skeletal muscle tissue 1106 nodes and 10555 edges (Supplementary Data 4). In total, these networks

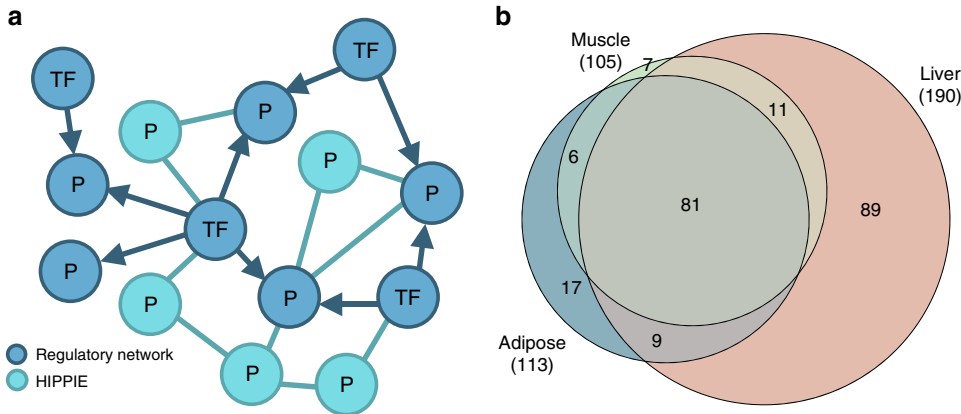

**Fig. 3** Network construction and disease module identification. **a** Tissue-specific networks were constructed by integrating transcriptional regulatory networks consisting of interactions between transcription factors and genes (blue nodes) and PPI networks including interactions among proteins (turquoise nodes). For each tissue, the integrated network was built starting from the high-evidence associations from the regulatory network and extended with high-score interactions from the PPI network. **b** Venn diagram of shared MetSyn genes among the three tissue-specific networks. See also Supplementary Data 4

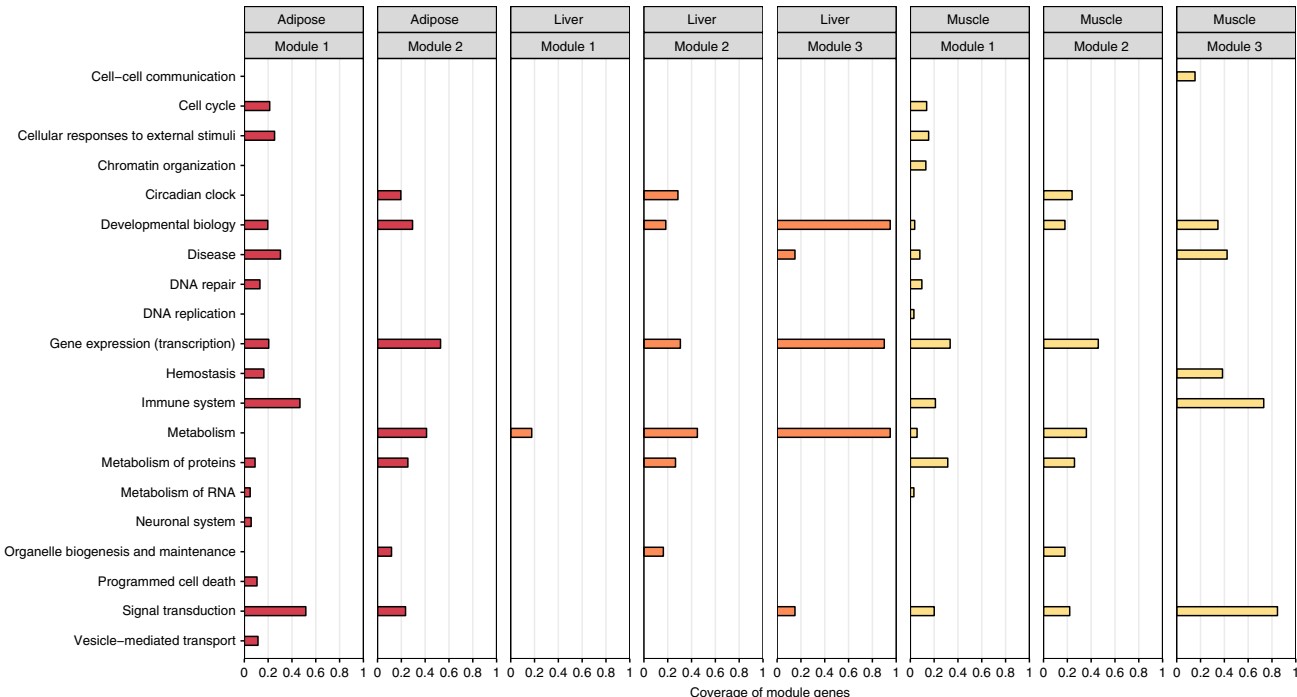

**Fig. 4** Functional annotation of the disease modules. For each network and disease module the coverage of module genes by significantly enriched Reactome pathways, grouped according to the TopLevel pathway classification, is presented. See also Supplementary Data 5

include 220 MetSyn genes, of which 81 are shared among the three networks (Fig. 3b). The liver tissue has the highest number of MetSyn genes not present in the other networks.

To identify trait-relevant network subparts, we tested network modules for their overrepresentation in MetSyn genes. For both the liver and muscle network, three significant trait-related modules could be identified, whereas we found two significant modules for the adipose tissue. Reactome pathway enrichment analysis of the genes in the network modules further allowed to link biological functions to the tissue-specific MetSyn modules. For example, in the adipose tissue network, the most significantly enriched pathways of module 1 are related to cellular responses to external stimuli and immune function (Cellular responses to heat stress: adjusted $p$-value $9.65 \times 10^{-7}$; immune system: adjusted $p$-value $9.65 \times 10^{-7}$). Instead, pathways related to metabolism regulation

resulted enriched in module 2 (PPARA activates gene expression: adjusted $p$-value $1.25 \times 10^{-11}$; regulation of lipid metabolism by Peroxisome proliferator-activated receptor alpha (PPARalpha): adjusted $p$-value $1.82 \times 10^{-11}$). The list of all significant pathways for the three tissue-specific networks can be found in the Supplementary Data 5. An overview of the module-related biological functions in all networks was obtained using the Top Level Pathways from the reactome database (Fig. 4). Overall, the resulting pathways show an overlap across tissues, highlighting the overrepresentation of genes involved in signal transduction and gene expression. Moreover, our results suggest that the immune system plays a considerable role for MetSyn; across all three networks a module with a high number of immune-related genes and pathways was detected (Fig. 4 and Supplementary Data 5), in agreement with previous reports[27,28].

The contribution of the different data sources to the identification of MetSyn modules is described in Supplementary Note 2 and summarized in Supplementary Tables 2–5. Overall, we observed that the combination of text mining and GWAS results allows a more detailed disease characterization than the two data sources independently. For example, the inflammation-related MetSyn module in the adipose network would not have been identified if the analysis had been limited to the genes derived solely from GWAS.

**Drug repurposing.** To identify drugs potentially affecting Met-Syn pathways, we selected approved drugs from DrugBank[29] having a target in at least one of the three networks (Supplementary Fig. 3). The interplay of MetSyn modules and 183 drugs was evaluated by computing the proximity score as shown in Fig. 5 and described in Materials and Methods. The score is based on topological properties of the network and functional similarity of the proteins. The contribution of these two components is described in Supplementary Note 2 and summarized in Supplementary Fig. 4. Drugs with a significant score point toward a possible disease indication or drug side effect. For the adipose network, this analysis resulted in a list of 28 significant drugs, for the liver network 31 significant drugs were identified, while for the muscle network the analysis resulted in 50 significant drugs. (Supplementary Data 6).

To test the effectiveness of our approach, we checked if drugs with known indication for adiposity, which has a key role in leading the metabolic disturbances associated with MetSyn, were identified by our scoring system. Bezafibrate, clofibrate, fenofibrate, gemfibrozil, mifepristone, pioglitazone were used for the evaluation process and 3 of them (pioglitazone, mifepristone and fenofibrate) were significance considering a threshold of 95 % (Supplementary Note 3 and Supplementary Table 6). After lowering the significance threshold to 85%, all six drugs were significant (Supplementary Fig. 5).

After having obtained the preliminary list of significant predictions, we performed a filtering and prioritization analysis to identify the most promising repurposing candidates (Supplementary Fig. 6). First, to exclude drugs with undesirable side effects, we evaluated information about contraindications from the DrugCentral platform[30] (Supplementary Data 7). For the

adipose results we excluded seven drugs, while for liver and muscle 12 and 24 drugs were excluded, restricting the list of repurposing candidates to 21, 19, and 26 drugs, respectively (Supplementary Data 6). Second, we filtered those drugs by focusing on their targets that were investigated using data from the OpenTargets platform[31]. For the adipose results, among the targets of the 21 drugs without known MetSyn-related side effects, 10 (AR, EGFR, HDAC6, IKBKB, NR3C1, PGR, PPARA, PPARG, RXRG, and VDR) have already been investigated for therapeutic interventions related to MetSyn and therefore we excluded them from further analyses (Fig. 6). For liver and muscle, six (NR3C1, NR3C2, PPARA, PPARD, PPARG, RXRG) and eight targets (ADRB2, AR, EGFR, ESR1, IKBKB, NR3C1, PPARA, RXRG) were removed, respectively (Supplementary Fig. 7, Supplementary Fig. 8).

After this filtering procedure, we identified the following drugs as having a potential novel therapeutic application for MetSyn: adapalene, afatinib, alitretinoin, belinostat, bosutinib, crizotinib, dequalinium, doconexent, erlotinib, ibrutinib, lapatinib, nintedanib, panobinostat, rucaparib, ruxolitinib, tamibarotene, tofacitinib (Table 1).

A final prioritization step was then carried out to evaluate if the tissue expression of the targets was concordant with the disease manifestations. Among the 18 targets of the drugs, only bruton tyrosine kinase (BTK), the target of ibrutinib, and nuclear receptor subfamily 1 group I member 2 (NR1I2), the target of erlotinib, showed a tissue-specific expression relevant for MetSyn. According to the Human Protein Atlas[32], GTEx[26] and FAN-TOM5[24] databases, *BTK* gene expression is consistently enhanced in immune-related tissues, and *NR1I2* expression is enriched in liver, while the other targets did not show any relevant tissue-specificity (Supplementary Tables 7, 8, and 9).

NR1I2 is a nuclear receptor that regulates hepatic detoxification, and is involved in glucose and lipid metabolism. Recent studies indicate that an activation of the protein could contribute to the development of MetSyn and diabetes[33]. Since erlotinib is an agonist of NR1I2, we concluded that the significance of the proximity score in the liver network could be explained by this finding. On the other hand, the BTK inhibitor ibrutinib is currently FDA-approved for the treatment of B cell cancers and the chronic graft-versus-host disease[34] while ongoing clinical

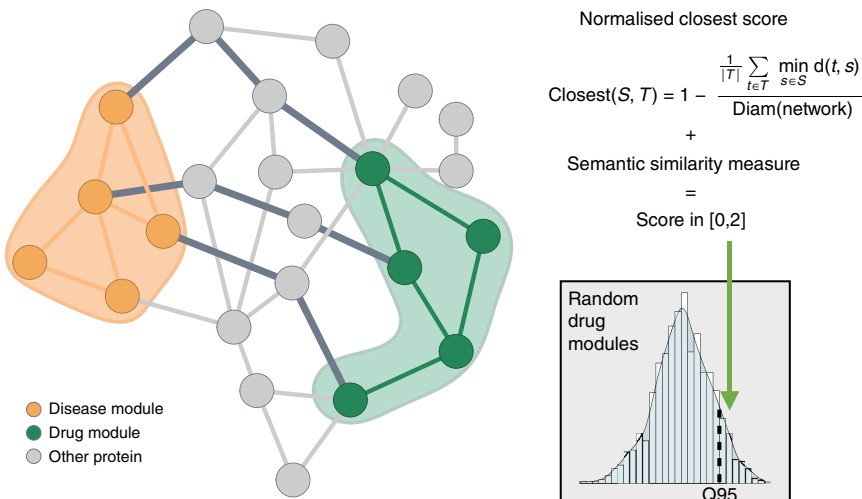

**Fig. 5** Illustration of the score calculation to evaluate the drug-disease interplay. The score combines network distances and functional similarity between proteins in the drug module (green nodes) and proteins in the disease module (orange nodes). The network score assesses the shortest path lengths connecting each protein of the drug module to the nearest protein in the disease module (dark gray edges) while the semantic similarity measure evaluates the functional similarity between the modules. To test the score significance, the distance between drug and disease modules is compared to a reference distribution of scores computed with drug modules randomly chosen from the network

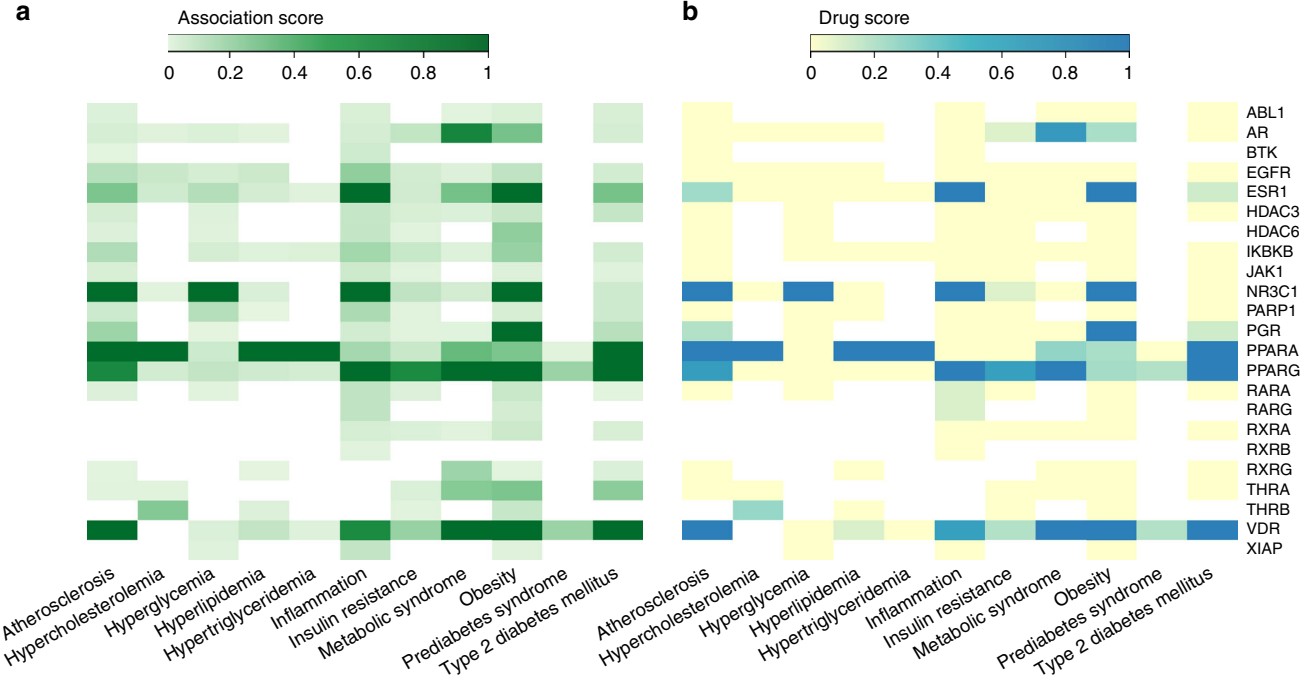

**Fig. 6** Association between the active drug targets in the adipose network and MetSyn-related traits based on the scores provided by OpenTargets. **a** Heatmap of total association score and **b** heatmap of association score based on ChEMBL information about drugs approved for marketing by FDA or under evaluation in clinical trials. Source data are provided as a Source Data file

trials evaluate the use of BTK inhibitors in autoimmune diseases, such as multiple sclerosis (ClinicalTrials.gov Identifier: NCT02975349) and rheumatoid arthritis (ClinicalTrials.gov Identifier: NCT03233230). Given the important role of inflammation in the alteration of adipose tissue biology in obese patients, we investigated the relationship between BTK and the immune system in obesity using public datasets. According to ImmGen mouse RNAseq data[35], the immune cell populations expressing high levels of Bruton tyrosine kinase transcripts are B cells and myeloid lineage cells such as neutrophils and macrophages (Supplementary Fig. 9). Interestingly, gene expression analysis of macrophages derived from adipose tissue of obese type II diabetic subjects (GSE54350)[36] showed higher *BTK* expression compared to macrophages of obese non diabetic subjects (Student's t-test p-value 0.026) (Fig. 7a). To further investigate *Btk* expression in obesity, we re-analyzed the adipose tissue transcriptome of a mouse model deficient in gpr120, a receptor for long-chain free fatty acids involved in nutrient sensing and body weight regulation (GSE32095). This mouse model, when fed with a high fat diet (HFD), was shown to develop obesity, insulin resistance, increased adipocyte size, and increased expression of macrophage markers[37]. Interestingly, we observed that these changes are coupled with an increased *Btk* expression in the adipose tissue, indicating the presence of an association between the pathophysiological changes observed in obesity and the increased expression of *Btk* in adipose tissue. In addition, the estimated composition of the adipose tissue-infiltrating immune cells in of the HFD-fed mouse, computed with CIBERSORT[38], revealed a significant increase in macrophages (Fig. 7d) compared with the mouse fed with a normal diet, underlining the prominent role of these immune cells in mediating the obese-related adipose tissue inflammation.

Since the macrophage-related inflammation in obese diabetic mice has been associated with inflammasome-dependent IL-1β production, we also evaluated the levels of Btk mRNA in adipose tissue of inflammasome-compromised mouse models

(GSE25205[38]). This dataset includes expression data derived from mice lacking Caspase-1, the enzymes that mediates the production of active IL-1β, or the inflammasome adaptor protein ASC. *Btk* expression was lower in white adipose tissue from HFD-fed Caspase-1 null mice than in wild type mice fed with the same diet (Fig. 7c), On the other hand, mice lacking the inflammasome adaptor protein ASC did not show a reduction in *Btk* expression in adipose tissue (Fig. 7c), suggesting a nonessential role of this protein for macrophage recruitment. This is in agreement with the results reported by Stienstra and colleagues, showing that HFD-fed ASC-null mice, despite displaying a healthier metabolic profile than HFD-fed wild-type mice, maintain an elevated macrophage infiltration in adipose tissue[38]. Overall, the analysis of these expression data shows an increased *Btk* expression in obese adipose tissue that is associated with the presence of macrophage infiltration. A similar pattern is observed by comparing BTK expression in human adipose tissue of obese and non-obese subjects, although the differences are less pronounced (Supplementary Fig. 10).

**In vivo validation of ibrutinib in a MetSyn model.** To validate the potential benefits of ibrutinib treatment for MetSyn, we established two zebrafish models of the disease by feeding 4 day postfertilization (dpf) larvae with high fat diets, containing either high fat cream (HFD[39]) or high cholesterol (HCD[40]), as shown in Fig. 8a. The efficiency of the two diets to induce fat accumulation was assessed with Oil Red O (ORO) (Fig. 8b) and Nile Red staining (Supplementary Fig. 11a) and we observed that both HFD and HCD were able to induce a significant increase of lipid accumulation in comparison with the standard diet (Fig. 8c). To evaluate the effects of the diets on the inflammatory response, we used transgenic lines with fluorescent macrophages—*tg(mpeg1: eGFP)gl22*[41] or fluorescent neutrophils—*tg(mpx:GFP)i114*[42]. Macrophages and neutrophils were clearly visible in live, anaesthetized larvae examined 7 dpf using an automated imaging

**Table 1 List of the drugs identified as possible repurposing candidates**

| DrugBank ID | Drug Name | Action | Target | Score | Module | Network |
|---|---|---|---|---|---|---|
| DB00210 | Adapalene | agonist | RARG | 1.6618 | 2 | Adipose |
| DB00210 | Adapalene | agonist | RXRB | 1.6925 | 2 | Adipose |
| DB00523 | Alitretinoin | agonist | RARG | 1.6661 | 2 | Adipose |
| DB04209 | Dequalinium | antagonist, inhibitor | XIAP | 1.7909 | 1 | Adipose |
| DB03756 | Doconexent | activator | RXRA | 1.7103 | 2 | Adipose |
| DB03756 | Doconexent | activator | RXRB | 1.6660 | 2 | Adipose |
| DB09053 | Ibrutinib | inhibitor | BTK | 1.6147 | 1 | Adipose |
| DB12332 | Rucaparib | antagonist | PARP1 | 1.6678 | 2 | Adipose |
| DB08877 | Ruxolitinib | inhibitor | JAK1 | 1.7721 | 1 | Adipose |
| DB04942 | Tamibarotene | agonist | RARA | 1.7222 | 2 | Adipose |
| DB00210 | Adapalene | agonist | RXRB | 1.6871 | 2 | Liver |
| DB03756 | Doconexent | activator | RXRA | 1.7047 | 2 | Liver |
| DB03756 | Doconexent | activator | RXRA | 1.4948 | 3 | Liver |
| DB03756 | Doconexent | activator | RXRB | 1.6819 | 2 | Liver |
| DB03756 | Doconexent | activator | RXRB | 1.5372 | 3 | Liver |
| DB00530 | Erlotinib | agonist | NR1I2 | 1.6684 | 2 | Liver |
| DB09079 | Nintedanib | inhibitor | FGFR3 | 1.4473 | 2 | Liver |
| DB04942 | Tamibarotene | agonist | RARA | 1.6667 | 2 | Liver |
| DB00210 | Adapalene | agonist | RXRB | 1.6759 | 2 | Muscle |
| DB00210 | Adapalene | agonist | RARG | 1.6676 | 2 | Muscle |
| DB08916 | Afatinib | inhibitor | ERBB2 | 1.6765 | 3 | Muscle |
| DB00523 | Alitretinoin | agonist | RARG | 1.7756 | 1 | Muscle |
| DB00523 | Alitretinoin | agonist | RARG | 1.6643 | 2 | Muscle |
| DB00523 | Alitretinoin | agonist | RXRB | 1.6620 | 2 | Muscle |
| DB00523 | Alitretinoin | agonist | RARA | 1.6810 | 2 | Muscle |
| DB05015 | Belinostat | inhibitor | HDAC1 | 1.7540 | 1 | Muscle |
| DB05015 | Belinostat | inhibitor | HDAC2 | 1.7615 | 1 | Muscle |
| DB05015 | Belinostat | inhibitor | HDAC4 | 1.8027 | 1 | Muscle |
| DB06616 | Bosutinib | inhibitor | ABL1 | 1.6090 | 3 | Muscle |
| DB08865 | Crizotinib | inhibitor | MET | 1.7883 | 1 | Muscle |
| DB03756 | Doconexent | activator | RXRA | 1.7134 | 2 | Muscle |
| DB01259 | Lapatinib | antagonist | ERBB2 | 1.8184 | 1 | Muscle |
| DB06603 | Panobinostat | inhibitor | HDAC3 | 1.6110 | 2 | Muscle |
| DB08877 | Ruxolitinib | inhibitor | JAK1 | 1.6745 | 3 | Muscle |
| DB08877 | Ruxolitinib | inhibitor | JAK2 | 1.7140 | 3 | Muscle |
| DB04942 | Tamibarotene | agonist | RARA | 1.6708 | 2 | Muscle |
| DB08895 | Tofacitinib | antagonist | JAK1 | 1.7100 | 3 | Muscle |
| DB08895 | Tofacitinib | inhibitor | JAK2 | 1.6920 | 3 | Muscle |

system (Operetta, Perkin Elmer) (Fig. 8d, Supplementary Fig. 11b, c). The number of fluorescent macrophages in the total body was significantly increased in larvae fed with HFD (Fig. 8e) whereas in cholesterol-fed larvae the increase was significant only in the region of head and yolk (Supplementary Fig. 11c, d). Moreover, we observed that macrophages in the two high fat diet conditions were morphologically different from those in larvae fed with a standard diet and appeared less dendritic (Fig. 8d, lower panels). On the other hand, there was no increase in the number of neutrophils (mpx:GFP + cells) with any of the two high fat diets (Supplementary Fig. 11e). The analysis of gene expression for key regulators of lipid metabolism (srebf1) and inflammatory responses (il1β) showed a significant increase in the expression of these markers in larvae fed with HFD (Fig. 8f-g), while the analysis of btk gene expression revealed that both diets resulted in a slight, although not significant, increase of btk expression (Fig. 8h). We then evaluated the effect of the Btk inhibitor ibrutinib on fat accumulation and inflammatory responses. Larvae were treated with 5 μM ibrutinib for 30 min, followed by HFD, HCD or control diet as shown in Fig. 9a. Treatment of HFD-fed zebrafish larvae with ibrutinib was able to significantly diminish the expression of srebf1 (Fig. 9b), and this reduction was coupled with a slight, albeit not significant, contraction of the number of larvae with high fat deposition (Supplementary Fig. 12a). In addition, the number of macrophages in HFD and HCD-fed larvae was significantly reduced by ibrutinib

treatment (Fig. 9c, d, Supplementary Fig. 12b) and the dendritic morphology seemed restored (Fig. 9c, lower panels). Moreover, the increase of the expression of the inflammatory cytokine il1β observed in HFD zebrafish larvae, was prevented by ibrutinib treatment (Fig. 9e).

## Discussion

The computational pipeline herein proposed depicts a systems biology approach to study the biological processes involved in MetSyn and to determine potential new pharmaceutical treatments via drug repurposing. Given their high interconnectivity and the multifactorial etiology, the metabolic disorders related to MetSyn are particularly suited to a system-level analysis that integrates multiple data types[43].

Although in this study we focused on MetSyn, the proposed method can be applied to other diseases. The only requirement is a list of disease genes that can be derived from many sources (e.g. from GWAS results or gene/protein expression profiles). Even in the case of Mendelian diseases, for which repurposing is an important opportunity to identify treatment strategies[44,45], the pipeline can be applied using information about affected pathways as input. Moreover, the network can be adapted for the specific requirements of the study. For example, if a disease-specific gene regulatory network is available, this can replace the GTEx-derived, tissue-specific networks selected here. On the other hand, the method can also provide useful insights for a

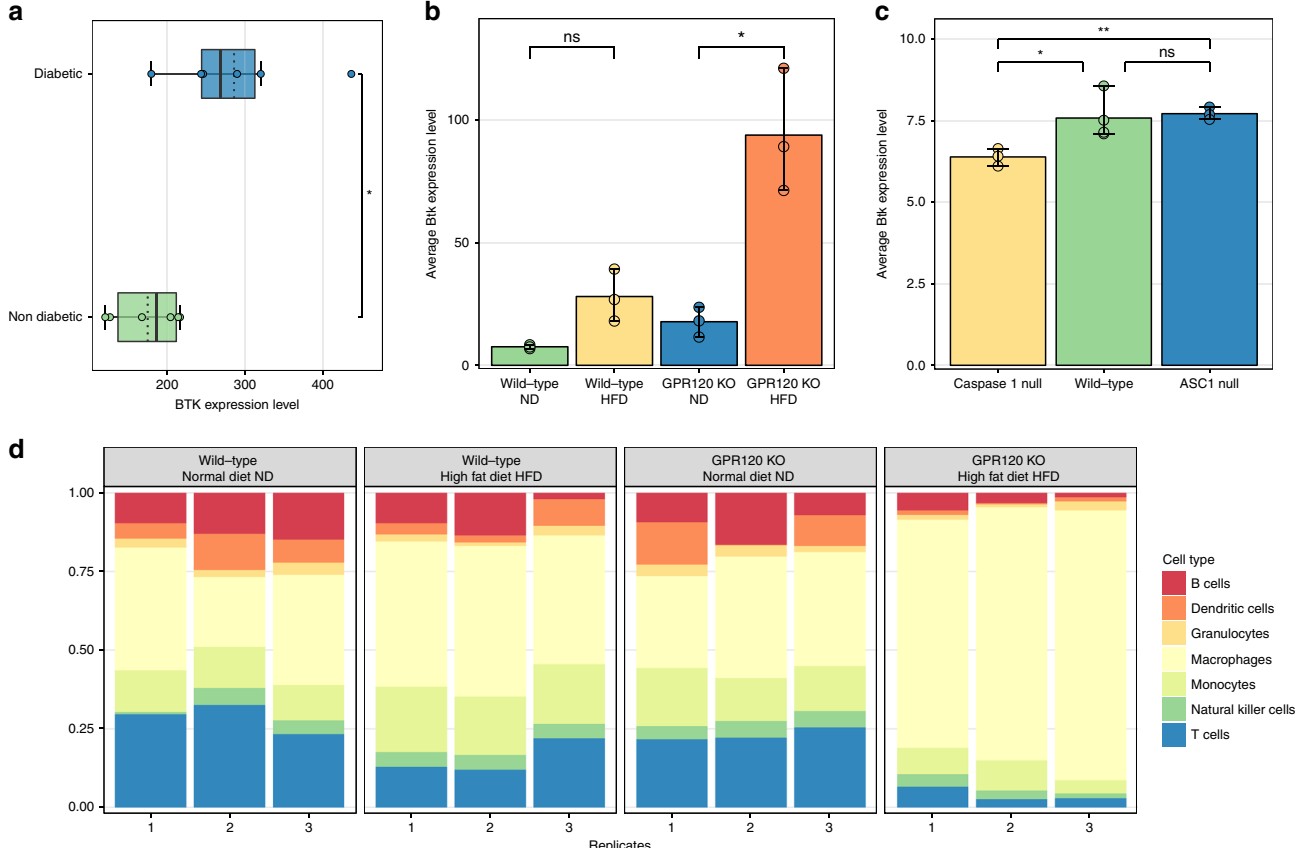

**Fig. 7** *BTK* expression in public datasets. **a** Boxplots showing *BTK* gene expression in macrophages of diabetic and non-diabetic subjects. The points represent the single values while the black tick lines indicate the median values and the dotted lines indicate the mean values. **b** Bar charts showing the gene expression level of *Btk* in adipose tissue for wild type and GPR120 KO mice fed with normal diet (ND) or high fat diet (HFD). The charts represent the mean of $n = 3$ replicates. The error bars indicate the min-max interval. **c** Bar charts showing the gene expression level of *Btk* in adipose tissue for wild-type, Caspase 1 null and ASC1 null mice. The charts represent the mean of $n = 3$ replicates for Caspase 1 null and ASC1 null mice and $n = 4$ for wild type mice. The error bars indicate the min-max interval. **d** Estimated relative fraction of different immune cells in adipose tissue calculated via Cibersort for wild type and GPR120 KO mice fed with normal diet (ND) or high fat diet (HFD) in adipose tissue. In a-c, statistical significance is denoted as follows: ns: not significant (*p*-value > 0.05), *:0.01 < *p*-value < = 0.05, **:0.01 < = *p*-value < 0.001 (Student's *t* test). Source data are provided as a Source Data file

specific drug of interest once network modules for various diseases are defined.

To identify MetSyn genes, we exploited GWAS and text mining results. The use of genetic evidence has been demonstrated as a powerful resource to support drug discovery, both for pointing out disease-related pathways and for the identification of new drug targets[46]. Indeed, the targets of many drugs approved before the GWAS era are located in GWAS risk loci, thus supporting further exploration of this data[47,48]. Moreover, the integration of GWAS results with literature findings allowed us to take into account the knowledge derived from additional sources, such as functional studies, and thus to have a more comprehensive view of the pathways involved.

The tissue-specific integrated networks were generated by merging a protein–protein interaction network and a transcriptional regulatory network. While the first describes known associations between proteins, such as the formation of protein complexes or kinase-substrate interactions, regulatory networks are fundamental to take into account the regulation exerted by transcription factors on gene expression. This is of particular importance when studying complex traits, because the regulation of gene expression in a tissue-specific manner is a key element in determining the pathological phenotype[26,49]. Moreover, it has been shown that several GWAS traits show higher connectivity in regulatory networks than in other types of networks[23], further

supporting our choice of including regulatory interactions in the background network.

A key point of the performed network analysis has been the definition of a proximity score that attempts to efficiently connect drug modules and MetSyn modules. In addition to the network-based distance between MetSyn and drug genes, our scoring system takes into account the similarity of biological gene functions. This approach increases the ability to identify potentially effective new therapeutic strategies because it adds direct biological knowledge to the topological information derived from the network.

The results obtained from the analysis of the adipose network remark the key role exerted by inflammation in obesity and suggest the adoption of anti-inflammatory therapies. This is in agreement with a growing body of literature supporting the use of anti-inflammatory agents to treat the chronic low-grade inflammation accompanying metabolic-related pathological conditions[50,51]. In particular, our results suggest ibrutinib as the most promising candidate for drug repurposing. This finding is based on the integration of genetic and literature data that allowed us to identify immune-related pathways as significantly enriched in disease genes. It is worth noting that the absence of text mining genes would have hampered the identification of ibrutinib.

Ibrutinib is a small molecule that inhibits BTK, a protein well known for its essential role in B cell development and

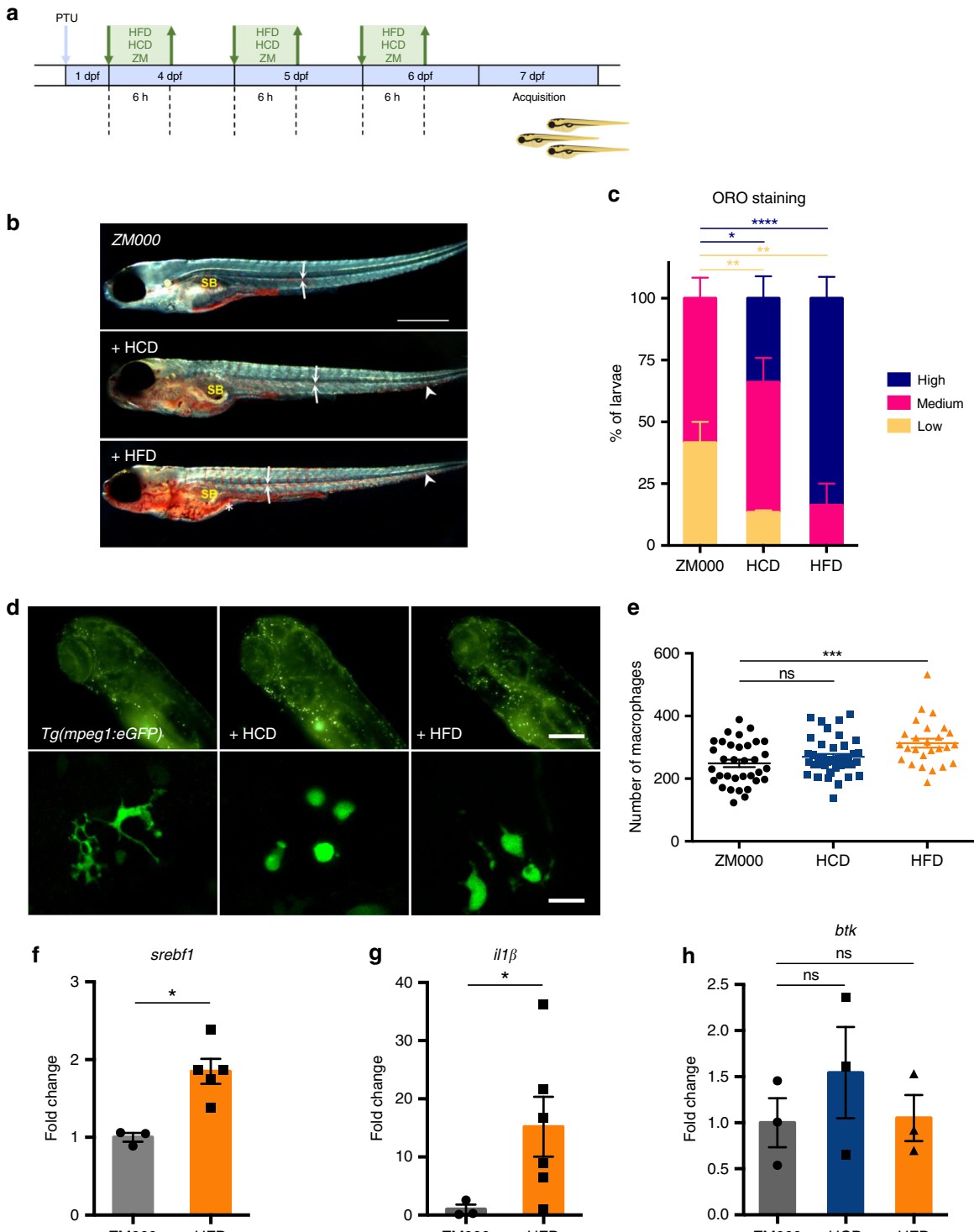

**Fig. 8** High fat diet induces lipid and macrophage accumulation in zebrafish. **a** Schematic representation of the protocol used for the experiments. Zebrafish larvae were fed with high fat (HFD), high cholesterol (HCD) or standard (ZM) diet for 6 h each day from day 4 to day 6 post fertilization (dpf). PTU: 1-phenyl 2-thiourea. **b** 7 dpf *Casper* zebrafish larvae, fixed and stained with Oil Red O: representative low (i), medium (ii), and high (iii) staining levels un larvae fed with the indicated diet are shown. SB: swim bladder. Arrows point to lipids in blood vessels; arrowheads point to lipid droplets in the tail region; asterisks indicate the intestine. Dark-field images, calibration bar: 500 µm. **c** Percentage of larvae with high, medium and low lipid accumulation for the three indicated diets. Data are pooled from two experiments ($n > 12$) and the charts show the mean ± SEM. Blue asterisks refer to statistics of low stained larvae, red asterisks refer to statistics of high stained larvae. ****$p$-value < 0.0001, **$p$-value < 0.01, *$p$-value < 0.05 (two-way ANOVA test). **d** Representative fluorescent images of the head + yolk regions of larvae obtained using the Operetta system (i–iii—calibration bar: 100 µm) and the confocal microscope (i'–iii"—calibration bar: 20 µm). **e** Quantification of macrophages for the three considered diets. **f**–**h** Quantitative PCR analysis of the expression of sterol regulatory element binding transcription factor 1 (*srebf1*), interleukin-1 beta (*il1β*) and bruton tyrosine kinase (*btk*) in larvae fed and treated as indicated. In **e** data are pooled from three or more experiments ($n >= 26$) and **$p$-value < 0.01, *$p$-value < 0.05 (Mann–Whitney test). Source data are provided as a Source Data file

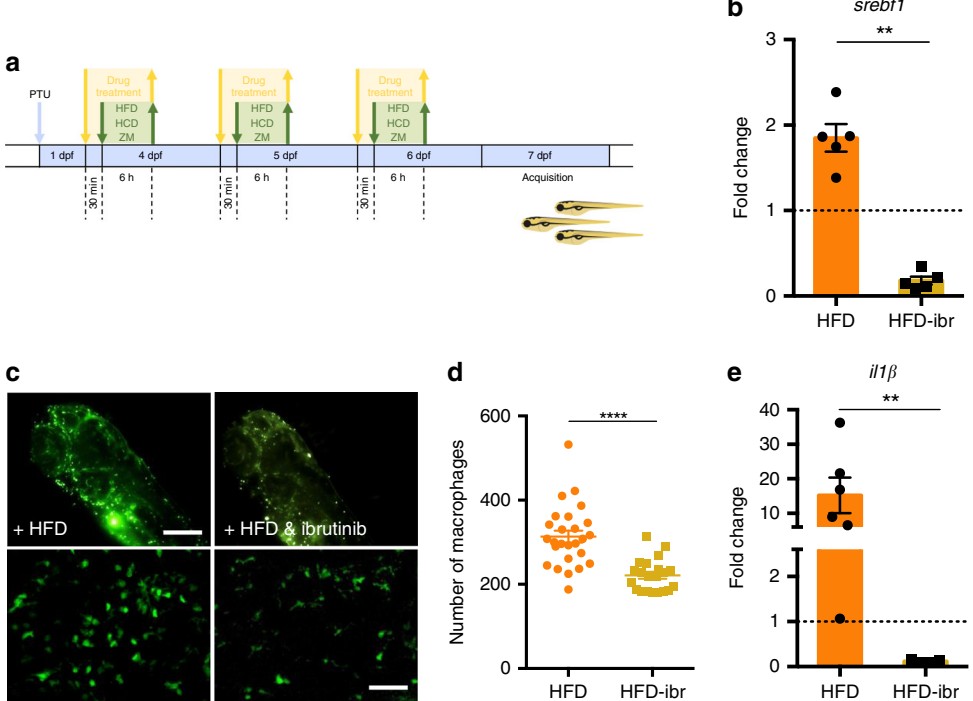

**Fig. 9** High fat diet effects can be prevented by ibrutinib treatment. **a** Schematic representation of the protocol used for ibrutinib treatment. Zebrafish larvae were treated with 5 µM ibrutinib 30 min before and during each feeding period. **b** Quantitative PCR analysis of the expression of *srebf1* in larvae fed and treated as indicated. The error bars show the standard error of the mean (SEM); **$p$-value < 0.01, $n = 5$ (Mann–Whitney test). **c** Representative images of the macrophages from the head + yolk regions of larvae obtained using the Operetta without (i) and with treatment (ii). Calibration bar: 100 µm. In i′-ii″ the macrophages are shown at higher magnification (calibration bars: 50 µm). **d** Quantification of fluorescent macrophages in HFD-fed larvae without and with ibrutinib treatment. Data are pooled from three or more experiments ($n > = 21$) and the charts show the mean ± the standard error of the mean (SEM). **$p$-value < 0.01 (Student's $t$-test). **e** Quantitative PCR analysis of interleukin-1β (*il1β*) expression in larvae fed and treated as indicated. **$p$-value < 0.01 (Mann–Whitney test). Source data are provided as a Source Data file

maturation[52]. In addition to B cells, *BTK* is also expressed by macrophages, known key players in the development of the obesity-related chronic inflammation and insulin resistance. At the molecular level, BTK is involved in the regulation of macrophage Toll-like receptor-mediated immune response[53,54] and is essential for the activation of the NLRP3 inflammasome and IL-1ß production[55]. Importantly, NLRP3 activity has been linked to obesity and insulin resistance both in human and mouse studies[56]. In addition to macrophages, B cells themselves have been implicated in adipose tissue inflammation and insulin resistance[57,58], providing additional support for BTK involvement in obesity-related inflammation.

In vivo zebrafish experiments support the potential of ibrutinib in reducing obesity-related inflammation. Indeed, in our zebrafish models of high fat diet-induced inflammation, the number of macrophages in the yolk and in the head region was reduced by ibrutinib administration. The reduction of macrophage number was accompanied by a diminished expression of molecular markers of lipid metabolism and inflammation, suggesting that ibrutinib may have long-term effects on lipid accumulation and associated inflammation.

Overall, this work describes a methodology based on the integration of genetic and expression data with previous biological knowledge, which enables the identification of drug repurposing candidates for complex diseases. Currently, this framework allows to test only drugs with a known target. A future extension could provide a method to also investigate compounds without a defined mechanism of action. Moreover, further developments may consider a refined fine mapping of the GWAS casual variants, for example by taking into account epigenomic

annotations. The application of the computational pipeline to MetSyn led us to identify the inhibition of Btk by ibrutinib as a promising repurposing strategy. Additional experiments are warranted to further investigate the effect of BTK inhibitors in obesity and the possible benefits for patients with metabolic syndrome.

## Methods
**List of genes associated with MetSyn**. To establish a list of genes associated with MetSyn, we used three different sources: GWAS catalog, GWAS summary statistics and text mining, as detailed below.

**Data-mining of the GWAS catalog**. Results of published genome-wide association studies were obtained from the NHGRI-EBI GWAS catalog (Ensembl release version E93, downloaded on 8 October 2018)[20]. MetSyn-related traits were manually selected among all those available, and the results were filtered for SNPs with an association $p$-value < 5E-08 (Supplementary Data 1). The extracted SNPs were mapped to official gene symbols based on their genomic location. All genes located in the genomic interval were considered. As reference genes, we used the RefSeq genes, downloaded from the UCSC genome browser using the table browser tool (human genome assembly: Dec 2013/HG38, downloaded on 11 October 2018 from: http://genome.ucsc.edu/index.html). Genes not assigned to chromosomes 1 to 22 were removed and the different transcriptional variants of one gene (isoforms) were merged by considering the minimal starting and the maximal ending position as new range of the gene. Moreover, the gene region was extended 110 kb upstream and 40 kb downstream of the transcript boundaries, following the approach used by MAGENTA[59]. The mapping procedure was executed with the R package GenomicRanges.

**SNP functional annotation**. The SNPs obtained from the GWAS catalog were annotated for their position within genes using the R package VariantAnnotation and TxDb.Hsapiens.UCSC.hg19.knownGene as annotation object. To evaluate the enrichment of the GWAS SNPs in regulatory regions, we used the chromatin state annotations from the NIH Roadmap Epigenomics project[60]. Specifically, the 18-

state models for adipose nuclei (E063), liver (E066) and skeletal muscle female (E108) were downloaded from https://egg2.wustl.edu/roadmap/web_portal/ and the overlap of the GWAS SNP locations with regulatory regions was computed using the R package GenomicRanges. For comparison, we downloaded the full set of HapMap CEU SNPs from the UCSC website (http://genome.ucsc.edu/) using Table Browser and annotated them in the same way we did for the GWAS SNPs. Two-sided Fisher's exact test was used to compare GWAS SNPs and HapMap SNPs.

**GWAS summary statistics**. A further resource for genetic factors associated with MetSyn is provided by GWAS summary statistics. Previously published results obtained by applying the PASCAL tool were used[61]. 15 GWASs connected to metabolic components were chosen (Supplementary Table 1) and genes with a p-value below the threshold of 5e-8 were selected.

**Text mining**. Additional MetSyn genes were identified using text mining of PubMed abstracts. The following MeSH terms were identified as being relevant to metabolic syndrome and its primary symptoms: metabolic syndrome x, hyper-glycemia, insulin resistance, hyperinsulinism, glucose intolerance, hypertension, obesity, abdominal, hypertriglyceridemia, hypercholesterolemia, waist cir-cumference, waist-hip ratio. The search was limited using the tags [Majr:NoExp], to restrict to articles having major focus on the searched MeSH terms (and no automatic inclusion of child terms of the searched term), and english[language] to restrict the search to English abstracts. In addition, we complemented the MeSH search with a keyword search using the PubMed [TIAB] tag. The following search terms were used: metabolic syndrome, hyperglycem*, insulin resistan*, hyper-insulin*, glucose intoleran*, hypertension, abdominal obesity, central obesity, hypertriglyceridemia, high triglycerides, hypercholesterolemia, high cholesterol, waist circumference, waist-hip ratio, waist-to-hip ratio. To limit the results to the most relevant articles, the keyword search results were filtered as follows. First, we removed those articles already annotated with MeSH terms, because either the major topic of the article was not considered MetSyn related by the MeSH reviewers, or the articles were already captured by our MeSH search. The remaining articles were further reduced to those waiting for MeSH annotation according to the MedlineCitation Status in PubMed (In-Data-Review, In-Process, Publisher), to cover the recent literature not yet included in the MeSH indexing.

Before performing the gene tagging, we removed those articles already present in our GWAS catalog results.

The genes mentioned in the titles and abstracts of the selected articles of both search strategies were annotated using the PubTator gene annotation[62], and filtered for human genes using the R package org.Hs.eg.db. PubMed was accessed on 24 October 2018 and PubTator annotation was downloaded on 25 October 2018.

**Combining the data-driven and text mining approach**. The approach we fol-lowed is based on genomic regions and thus includes also genes that are not causative risk factors for metabolic syndrome. Furthermore, the possibility of including false-positive results from the text mining approach cannot be discarded. To deal with these two limitations, gene set enrichment analysis for the genes identified by the data-driven approach was carried out and genes were prioritized based on their biological function. The enrichment analysis was performed using the enrichr tool[22], accessed through the RESTful API on October 12,2018. We selected the following databases: GO biological processes, KEGG, WikiPathways, Reactome, Biocarta, Humancyc, NCI-Nature, and Panther. The analysis resulted in 47 significant gene sets (BH adj. p-value < 0.05).

**Constructing tissue-specific background networks**. Integrated tissue-specific networks were constructed by combining two types of networks: transcriptional regulatory networks composed of interactions between transcription factors and the regulated genes, and human protein–protein interaction networks. We consider adipose tissue, liver tissue and skeletal muscle tissue as the three tissues mostly affected by MetSyn.

Regulatory networks were downloaded from http://regulatorycircuits.org/download.html as presented in[23]. These tissue-specific gene regulatory networks were inferred by combining transcription factor sequence motifs with activity data for promoters and enhancers from the FANTOM5 project[24]. Among the available individual networks, adipose_tissue_adult, liver_adult and skeletal_muscle_adult were selected. Based on the activity scores, edge weights in the range of [0, 2] are provided. To filter for interactions with high evidence scores, we chose a cut-off value for these edge weights of 0.4. This value is based on the considerations, that (a) a threshold over 0.5 results in networks without/with only few nodes, (b) a threshold beneath 0.1 rises the possibility of false positives steadily as the distribution of edge weights is highly skewed, and (c) the threshold of 0.4 is the maximal value which secures that at least 25% of the TFs and PRs of the resulting network are also expressed in the tissue-specific corresponding network from HIPPIE as additional source for protein–protein interactions.

Protein–protein interactions (PPI) were obtained from HIPPIE (v2.0)[25]. HIPPIE is a comprehensive database combining protein interactions from different sources. Furthermore, a confidence score for each interaction is provided. This score ranges in [0, 1] and reflects the reliability of the interaction based on the

number and the quality of the experimental technique, the number of studies mentioning the interaction, and the number of non-human organisms in which the interaction was reproduced. To include only the most reliable interactions, the cut-off value of 0.73 was chosen considering that the curators of HIPPIE refer to this value for high evidence interactions. Following[25], tissue-specific PPIs were created using tissue expression RNA-Seq data from GTEx, while a gene was considered tissue-relevant if it showed an RPKM ≥ 1 in the given tissue. To generate the adipose tissue network, we combined the data obtained from subcutaneous and visceral adipose tissue.

After this preprocessing analysis, the regulatory circuits were extended with high evidence interactions from the respective tissue-specific PPI. Relations for nodes in the gene regulatory networks as well as their first neighbors were included during this process.

**Drug data**. To evaluate the drug effect on MetSyn we combined drug target information and drug expression profiles.

**Target information**. Information about drugs, their indication, their stage of development (e.g. approved, experimental, withdrawn) and their target was obtained from the public database DrugBank[29] (version 5.1.1, release date: 2018-07-03, download date 11 September 2018). Drugs were retained if they were annotated to have a target gene and if they had an approved status. We further restricted our analysis to pharmacological active drug-target interactions. All target proteins were mapped and annotated using Entrez IDs and official gene symbols. In total, we obtained 3814 drug-target interactions for 1482 distinct drugs and 705 distinct targets.

**Drug expression profiles**. The identified drug-target relations were extended by including knowledge about the gene expression profile related to the drug, obtained from the Library of Integrated Cellular Signatures (LINCS)[11], which provides gene expression profiles obtained by analyzing cellular responses (cellular sig-natures) across different cell-lines in response to a range of perturbations, including also single drug perturbations. We accessed the data using the RESTful API (https://clue.io/) in October 2018 and for each drug the 100 most up- and down-regulated genes were retrieved, relying on high quality signatures (is_gold = 1). If for a certain drug more than one signature was available, we selected the one with the highest signature strength parameter (distil_ss).

**Network-based MetSyn modules**. Trait-relevant network modules were detected using the walktrap algorithm[63] (implemented in the R package igraph) in com-bination with an overrepresentation test. The walktrap algorithm identifies network communities based on the concept that random walks of a short length tend to stay in the same network area (identified as module). The algorithm was run using the default parameters. The enrichment in MetSyn genes was tested using one-sided Fisher's exact test (p-value < 0.05). The communities significantly enriched in MetSyn genes were tested for their biological functionality using pathway enrich-ment analysis (R package reactomePA).

**Network-based drug modules**. Network-based drug modules were generated by mapping drug profiles to the networks and connecting the mapped proteins. Starting from the drug target, the list of signature proteins was filtered to extract those having a medium to high semantic similarity with the target protein using the R package GoSemSim (Wang method[64], cut-off value 0.5). The network-based drug module was then formed by the drug target, the selected subset of drug signature proteins and the shortest paths connecting them.

**Proximity score**. A proximity score was defined to quantify the interplay between a drug profile and a MetSyn module. This score combines the network-based distance between a drug- and a MetSyn-module, and the semantic similarity of the two modules. The network-based distance was calculated using the closest distance introduced in[15], where it has been shown to outperform other distance measures. This measurement represents the average shortest path length between the drug module genes and their nearest disease proteins in the network:[15]

$$d_c = \frac{1}{|T|} \sum_{t \in T} \min_{s \in S} d(s, t), \tag{1}$$

where $T$ is the drug module, $S$ the disease module and $d(s,t)$ the shortest distance between two nodes $s$ and $t$. Normalizing this measurement with the diameter of the network and considering the linear transformation $1-d_{c, norm}$ defines a score in [0,1]. To include knowledge about the biological function of the drug and disease proteins, their GO annotation restricted to biological processes was used to cal-culate a similarity score in [0,1]. Wang's method[64] combined with the Best-Match Average strategy was used as implemented in the R package GoSemSim. Summing the two above measurements led to our final score in [0,2].

To assess the significance of the results, a reference score distribution corresponding to the expected scores for random sets of drug proteins was created. The construction of the random module follows the strategy to build drug modules described above by selecting first a target protein falling in the same degree bin as

the original target, and by then selecting signature genes keeping the internal distances of the original module. Finally, we use the shortest paths between the target and signature genes to construct the random module. A drug resulting in a score higher than 95% of the reference distribution scores was considered significant.

**Filtering and prioritization of candidate repurposing drugs**. We retrieved data about the targets of the repurposing candidates using the Open Targets Platform[31] REST API (accessed November 2018) to extract known associations between the target genes and the list of traits associated to MetSyn. Targets with at least one association score ≥ 0.2 were excluded from further considerations, because this indicates that the target has already been under investigation for therapeutic interventions related to MetSyn. Furthermore, we extracted the known side effects of the candidate drugs using the DrugCentral platform[30], accessed via http://drugcentral.org in November 2018, and excluded the drugs with contra-indications associated to MetSyn. A final prioritization step was carried out based on the tissue expression of the drug targets accessed using Human Protein Atlas[32], GTEx[26], and Fantom5[24].

**Analysis of *BTK* gene expression**. The mouse and human expression datasets described in this study are publicly available from NCBI GEO (https://www.ncbi.nlm.nih.gov/geo/) and EMBL-EBI ArrayExpress (https://www.ebi.ac.uk/arrayexpress/). From NCBI GEO we downloaded the Series_matrix files of the following datasets: GSE54350, GSE32095, GSE25205, and GSE27951, while the processed data of the E-MTAB-54 dataset was downloaded from EMBL-EBI ArrayExpress. The selected datasets were annotated using the R Bioconductor annotation packages corresponding to the microarray platform used in the respective study or the annotation file provided by NCBI and ArrayExpress. The probe signals were summarized at gene level considering the median. T-test was used to compare the mean of *BTK* transcript levels between different subgroups.

**Immune cell component estimation**. To estimate the abundances of immune cells in adipose tissue we used the online version of Cibersort[65] (accessed via https://cibersort.stanford.edu/), run with default parameters. Cibersort is software based on a deconvolution algorithm that allows estimating the abundances of immune cells from gene-expression data on the basis of previous knowledge about immune cell gene expression (immune signature). For the human datasets, we used the immune signature provided by Cibersort that contains 22 immune cell types, while for the mouse datasets, we used the immune signature provided in Chen et al.[66], consisting of 25 immune cell types. For visualization purposes, the mouse immune cells were grouped in seven main classes: Granulocytes (Mast Cells, Neutrophil Cells, Eosinophil Cells), B cells (B Cells Memory, B Cells Naïve, Plasma Cells), T cells (T Cells CD8 Actived, T Cells CD8 Naïve, T Cells CD8 Memory, T Cells CD4 Memory, T Cells CD4 Naive, T Cells CD4 Follicular, Th1 Cells, Th17 Cells, Th2 Cells, GammaDelta T Cells), Macrophages (M0 Macrophage, M1 Macrophage, M2 Macrophage), Monocytes (Monocyte), Natural Killer cells (NK Resting, NK Actived), and Dendritic cells (DC Actived, DC Immature).

**Animal rearing**. Zebrafish (*Danio rerio*) strains were raised and maintained in the Model Organism Facility (MOF) at Department of Cellular, Computational and Integrative Biology (CIBIO) – University of Trento under standard conditions[67]. The transgenic zebrafish lines *tg(mpeg1:eGFP)gl22*[41], *tg(mpx:GFP)i114*[42] and *Casper*[68] were used. All animal experiments were performed in accordance with European guidelines and regulations, and were approved by the ethic board of the University of Trento. Approval for breeding was granted by the local government (Città di Trento, C_L378/S022/103359/03.06.2015 to Università di Trento).

**Preparation of experimental diets**. To create cholesterol-enriched diet (HCD) for feeding, cholesterol (Sigma) was diluted in diethyl ether (Sigma) to obtain 10% solution. 400 μl of the solution was added to 0.5 g of standard zebrafish larval food (ZEBRAFEED, ZM000 ingredients: crude protein 63%, crude fat 14%, crude fiber 1.8%, crude ash 12%). Control diet was obtained adding 400μl of diethyl ether to 0.5 g of ZM000. The diets were left overnight under the chemical hood until the diethyl ether was evaporated. Before feeding larvae with diets, they were crumbled into fine particles using a small scoop. Preparation of diets was performed under sterile condition (adapted from Progatzky et al.[40]). High-fat diet (HFD) was created diluting 1:10 clotted cream (Devon Cream Company, ingredients: 55.0 g of fat, of which saturates 35.5 g, 2.2 g of carbohydrate, of which sugar 2.2, 1.6 g of protein, trace of salt) in E3 (5 mM NaCl, 0.17 mM KCl, 0.33 mM CaCl2 × 2H2O, 0.33 mM MgSO4 × 7H2O, 0.0002% methylene blue, pH 6.5) containing 200 μM 1-phenyl 2-thiourea (PTU, Sigma) (adapted from Schlegel and Stainier[39]). HFD was prepared daily, instead HCD and control diet (ZM000) were prepared once and stored at room temperature.

**Feeding of Zebrafish larvae**. Larvae were initially maintained at 28.5 °C at a maximum density of 50 larvae per Petri dish in E3 fish water, containing 0.0002% methylene blue (Sigma) as an antifungal agent. After 24 h, larvae were placed in E3 fish water, containing 200 μM PTU, to prevent accumulation of melanin and allow

fluorescent imaging of inflammatory cells. Before feeding, mpeg:GFP and mpx:GFP positive larvae were selected using a Leica MZ10F Stereomicroscope and were randomly assigned to the differed treatment groups. Sample size per treatment group was 15 zebrafish larvae, number of experiments was > 3. From 4 dpf to 6 dpf, larvae were placed in HFD or in E3 in which HCD or standard diet (ZM000) were added using a small scoop. Feeding was performed for 6 h per day at 28.5 °C. Total number of macrophages was counted at 18 h post feeding (at 7 dpf) (adapted from Progatzky et al.[40]).

**Lipid staining**. Oil Red O staining was performed on *Casper* zebrafish fixed larvae at 7 dpf as described by Progatzky et al.[40] and Riu et al.[69], with slight modifications. Briefly, larvae were fixed in 4% paraformaldehyde overnight at 4 °C. Fixed larvae were washed three times in PBS 1X for 5 min and incubated in 60% isopropanol for 30 min. Larvae were then incubated with freshly prepared Oil Red O staining solution (0.3% Oil Red O in 60% isopropanol) for 3 h. After staining, larvae were washed three times in 60% isopropanol for 5 min before being transferred in PBS ×1. Images were acquired at the stereomicroscope in darkfield mode.

Nile Red staining was performed on 7 dpf *Casper* zebrafish live larvae, as described by Ma et al.[70]. Larvae were incubated in 5 ml E3 fish water with 0.5 μg/ml Nile Red solution (1.25 mg/ml stock solution in acetone, diluted in E3 fish water) for 30 min in the dark at room temperature. Images were acquired at the confocal microscope.

**Automated count of macrophages and neutrophils**. Embryos at 7 dpf were manually arrayed into 384-well plates (Corning®) in 70 μl of E3 fish water. They were anesthetized with 0.2 mg/ml MS-222 (Sigma-Aldrich) and manually positioned on their side in the upper left corner along the diagonal of the well. Image acquisition was performed by Operetta High Content System (Perkin Elmer) of the HTS Facility at CIBIO Department (University of Trento). Region of interest was acquired in three fields of view, with a 10X objective (NA = 0.4) and included the head, the yolk and the tail. Images were acquired in Widefield mode with two channels: brightfield and 460-490 Ex/500-550Em for EGFP. Nine z-stacks with 20 μm steps were sufficient to obtain a representative image for each fish. Resulting maximum projection was analyzed using Harmony® software (Perkin Elmer). All measurements were taken from distinct samples.

Images were pre-processed by filtering and smoothing. Signal normalization was performed to reduce artifacts and increase the inter-plate comparability. Fish region was defined using the eGFP diffuse signal. Cell detection in the defined region was performed using a highly inclusive method based on the eGFP normalized signal (Supplementary fig. 11b). Multiple properties of the cells were calculated and macrophages or neutrophils were selected in the respective transgenic larvae by automated pattern recognition relying on supervised machine learning (PhenoLOGIC™ Perkin Elmer). Based on the selected training objects, a linear combination of properties is identified which best separates the training samples (Supplementary Fig. 11b). The number of objects per field of view was counted and summed to obtain the total number of cells per larva. The resulting cell segmentation and eGFP positive cell selection is shown in Supplementary Fig. 11b.

**Treatment with ibrutinib**. Treatment with ibrutinib was tested at different dosages (5–50 μM); as some toxicity was present at 50–20 μM (not shown), we used a dose of 5 μM. From 4 dpf to 6 dpf, zebrafish larvae were pre-treated for 30 min with 5 μM ibrutinib, followed by feeding for 6 h with HFD, HCD or control diet (ZM000). Pre-treatment and feeding were performed at 28.5 °C. Quantification of total number of macrophages was analyzed at 18 h post last day of feeding (at 7 dpf). For pre-treatment, ibrutinib was administered into E3 fish water containing PTU. Regarding feeding with HCD and control diet (ZM000), ibrutinib was administered into E3 fish water containing PTU, in which each diet was added; instead for HFD, ibrutinib was directly administered into this specific diet.

**RNA extraction, cDNA synthesis, and qPCR**. RNA extraction was performed using ReliaPrepTM RNA Tissue Miniprep System (Promega), according to the manufacturer's protocol. At the final step, RNA was eluted with 15 μl water to increase its concentration. Quantity and quality of RNA were assessed through NanoDrop 2000c (Thermo Scientific). 0.5 nanograms of total RNA was retro-transcribed to cDNA using SensiFAST™ cDNA Synthesis Kit (Bioline), according to the manufacturer's protocol. Quantitative PCR reaction mix was prepared using qPCRBIO SybGreen Mix (PCR Biosystems), according to the manufacturer's protocol. cDNA was diluted 1:4 with water. qPCR reaction was performed on a CFX96 Real-Time PCR Detection System (Bio-Rad) machine to amplify btk, il1b[70], srebf[70], and rps11 (housekeeping gene used as a reference) mRNA. All measurements were taken from distinct samples.

Primers were purchased from Eurofins Genomics: btk forward primer 5′-CCCA CGAGTATTGCGCTTCT-3′, btk reverse primer 5′-GACTTCAGGAGGTGACC AGC-3′; srebf1 forward primer 5′-CATCCACATGGCTCTGAGTG-3′, srebf1 reverse primer 5′-CTCATCCACAAAGAAGCGGT-3′; il1b forward primer 5′-AC ACCGAGCGCATCATTAAC-3′, il1b reverse primer 5′-TGCGTCAGTAGTGTTG GTCT-3′; rps11 forward primer 5′-ACAGAAATGCCCCTTCACTG-3′; rps11 reverse primer 5′-GCCTCTTCTCAAAACGGTTG-3′.

**Statistical analysis of zebrafish data**. Statistical analyses of zebrafish data were conducted using GraphPad Prism 6.0e software. Analysis of macrophage and neutrophil count was carried out applying two-tailed unpaired Student's *t*-test and values were presented showing the standard error of the mean (SEM). Analysis od Oil Red O staining data was conducted by applying the two-way ANOVA test with Bonferroni post-test and values were presented showing the SEM. Analysis of qPCR data was conducted applying the two-tailed Mann–Whitney test for unpaired sample and values were presented showing the standard error of the mean (SEM). Statistical values of *p*-value < 0.05 were considered significant: **** stands for *p*-value < 0.0001, *** stands for *p*-value < 0.001, ** stands for *p*-value < 0.01 and * stands for *p*-value < 0.05.

**Fluorescence imaging**. A fluorescence stereomicroscope (Leica MZ10F) was used for phenotype selection of *tg(mpeg1:eGFP)gl22* and *tg(mpx:GFP)* i114 larvae, using an eGFP filter. For live imaging, zebrafish larvae were anaesthetized in 0.2 mg/ml MS-222. Images were acquired through confocal laser scanning microscope (Leica, SP5; objectives: ×10/0.8, ×20/0.8). Live imaging was performed mounting 7 dpf larvae per each treatment in 1% low melting point agarose (Life Technologies) with a lateral orientation.

**Image preparation**. Images were prepared for publication using ImageJ 1.50i (Fiji) program.

**Reporting summary**. Further information on research design is available in the Nature Research Reporting Summary linked to this article.

## Data availability
The list of SNPs associated with metabolic syndrome analyzed in this study was obtained from the NHGRI-EBI GWAS catalog. The gene regulatory networks were retrieved from regulatory circuits portal. The protein–protein interaction network was downloaded from HIPPIE. The drug expression profiles were retrieved from the Library of Integrated Cellular Signatures (https://clue.io/) portal. The gene expression datasets were downloaded from NCBI GEO database and from EMBL-EBI ArrayExpress. The list of drugs and targets was downloaded from DrugBank. The data about the drug targets were downloaded from Open Targets platform. The data about the side effects were downloaded from Drug central platform. The data produced by the analyses in this manuscript are available within the article and its Supplementary Information files. The source data underlying Figs. 6a-b, 7a-d, 8c, 8e-h, 9b, 9d-e and Supplementary Figs. 7a-b, 8a-b, 10a-d, 11d-e, 12a-b are provided as a Source Data file.

## Code availability
The code to compute the proximity score defined in this article is available at https://www.cosbi.eu/fx/936294736.

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

## Acknowledgements

We would like to thank Michael Pancher of the HTS facility at CIBIO Department for help with the Perkin Elmer Operetta system and related software. The work carried out at COSBI was partially supported by the PAT (Provincia Autonoma di Trento).

## Author contributions

S.P. and C.P. conceived and designed the study; K.M. performed the computational analysis; S.P. performed the biological interpretation of the computational results; S.P., M.M., L.L., and P.B. contributed to the computational analysis; F.L., V.S. performed the zebrafish experiments; S.P., M.C.M., E.D., and C.P. supervised the study and provided guidance on data analysis; K.M. and S.P. wrote the paper; all authors reviewed the paper and approved the final version.

## Competing interests

The authors declare no competing interests.
