## [Peer Review File · Nature Communications]

Reviewers' comments:

Reviewer #1 (Remarks to the Author):

This paper entitled "Metabolic syndrome: identification of deregulated pathways and drug effects by network analysis" by Misselback K et al. shows network analysis of integration of metabolic data, genetic data, tissue array and drug-related databases. While this paper presents a novel approach and contains intriguing data, a couple of questions could be raised.

1. From extensive network analysis, Btk was chosen as a new target. However, it is not known or addressed where Btk have impacts, while some analysis data suggesting the role of Btk in metabolic inflammation or inflammasome activation have been presented. It is recommended to conduct some in vitro experiment (even if preliminary) to explore the immunological steps affected by Btk.
2. Ibrutinib was selected as a candidate for new drug repositioning. It is recommended to conduct experiments studying the effect of ibrutinib in metabolic inflammation/inflammasome activation in vitro and metabolic syndrome in vivo.
3. Some more discussion regarding other seven drugs in Table 1 besides ibrutinib will be helpful.

Reviewer #2 (Remarks to the Author):

In this manuscript, the authors describe a computational approach to finding new therapeutics for the metabolic syndrome through drug repurposing. Their proposed method is based on network analysis through integrating drug- and disease-related genes via various networks. The work led to a novel finding that ibrutinib is a potential new drug for the metabolic syndrome. The work certainly adds to the growing body of literature on computational drug repurposing through network-based analysis. The work is solid overall but could be further strengthened with regard to its main contributions to the technical advancement as this is mainly a methodology article in its current form. In addition, more lab work is certainly warranted for the further validation of their finding in practice. My specific comments can be found below:

1. The study provides some experimental evidence (in Fig 7) but as the authors concluded themselves "more experiments are needed to investigate ... the possible benefits for patients with metabolic syndrome." Thus I believe the major contribution in this work lies in their proposed computational method. From the technical perspective, what is the unique novelty in your method vs. the previous network-based approaches. Highlighting such differences would be helpful for those interested in the algorithmic development and improvement. The generalizability of the method should also be extensively discussed.
2. Because of the multi-step approach, I feel it is important for the authors to have evaluations of the derived data in each individual step. In other words, it is difficult to judge the data quality and completeness from the current description.
3. It makes sense to supplement GWAS with literature mining. However, I'd say that using simple search with MeSH terms is not ideal (e.g. new articles are not indexed with MeSH). Furthermore, obtaining gene information from the list of MeSH substance is problematic as many gene/protein concepts are absent from MeSH (this is a known problem for MeSH vocabulary). Therefore, I'd suggest the authors either manually tag the genes from their search results or find an automated gene tagger.

As can be seen in Figure 1, there exist a number of genes, unique to text mining, even after filtering in the enrichment analysis. Have they been somewhat looked at at all?

4. The authors stated the critical importance of combining data from different sources in various steps of their approach. e.g. combining GWAS with literature or merging PPI with transcriptional regulatory network. It would be helpful to tease out the effect of individual component for the final system prediction.

5. A known challenge in computational drug repurposing is selecting promising candidate drugs from the list of predictions made by an algorithm. Was your predication list manually examined and is this why only the results for the adipose network are investigated? Furthermore, how was the set of 8 drugs in table 1 identified and why the authors decided to further pursue Ibrutinib from the list (due to experimental data availability?) Not sure if it is applicable but other types of algorithm validation that are worth looking into might be the use of patient EHR data (e.g. Cheng et al., Nat Comm, 2018).

6. According to the article, only 3 out of 6 MetSyn drugs were found to be significant in their analysis, suggesting the sensitivity of their approach is 50%. what about the other three? can you please discuss this in the paper.

Minor:

- Having a brief discussion of the results from the other two networks would be nice.
- I noted in a few cases where data were downloaded in 2016 (e.g. GWAS, text mining). Since it's fall 2018 now, would it make sense to update and use the latest version for your analysis?
- Please replace comma with dot for the decimal scores in Table 1.

Reviewer #3 (Remarks to the Author):

The authors proposed a drug repurposing approach based on network construction and analysis to identify therapies for metabolic syndroms.

The paper is missing of a proper introduction on the background of the data used in the approach. For example, why did the authors used GWAS? The authors should claim what are the pros and the cons of the data used. I think the use of the GWAS catalog and the SNPs related to the Metabolic Syndromes to start the network construction is an important and innovative step. The authors should discuss more about it since this is the starting point to construct the network. The paper lacks comparisons about different approaches both toward the drug repurposing and network analysis metric (proximity score).

Examples:

- Cheng, Feixiong, et al Nature communications 9.1 (2018): 2691
- Vitali, F., Journal of biomedical informatics, 46(5), 876-881.

The novelty of the paper it is unclear.

The results are difficult to follow without reading the methods as a lot of details on the adopted procedures are missing.

- The authors identified the starting genes from the GWAS Catalog, another study (Supplementary Table 1) and text mining. In Fig 2a is shown that most of the genes came from the text-mining search. This type of evidence can be considered less reliable if compared to the other information. How the use of this type of evidence affect the results?
- How the tissues were included in the network? Where this information came from? The authors should make it clearer in the results as well
- Line 98: how the enrichment has been performed? Using Reactome?
- Line 147 – how the information on target have been integrated?
- Figure caption and number are missing.

The results should be more robust, for example the authors claimed (line 145) that in the adipose network 3 out of 6 MetSyn drugs were identified thanks to their approach. The authors should state how they identified the six drugs and which are these drugs.

The authors proposed to use a proximity score that combines network distance with semantic similarities, they should justify better their choice since other methods already exist for that aim

(e.g. Vella, D (2018). Scientific reports, 8(1), 5499)

The authors validate one drug candidate out of the 8 drugs identified as potential by analyzing the expression of its target in public gene expression data of obese patients.

This limits the impact of the paper. Additional comments on the other drugs identified should be provided. Also, there is no experiment in verifying the candidates.

Other minor comments:

- Line 73: GWAS reference missing

- The authors should provide the figures on the three-tissue network identified, and preferably their node and edges as supplementary files.

- In Computational framework overview: the authors should reference their approach steps with the number reported in Fig1.

We thank the reviewers for their time and effort spent reviewing our manuscript, and for their helpful comments and suggestions. In light of their reviews, we updated the computational workflow, included *in vivo* experiments to validate the main *in silico* finding and modified the manuscript accordingly. Point-by-point responses are reported below.

Reviewer #1 (Remarks to the Author):

This paper entitled “Metabolic syndrome: identification of deregulated pathways and drug effects by network analysis” by Misselback K et al. shows network analysis of integration of metabolic data, genetic data, tissue array and drug-related databases. While this paper presents a novel approach and contains intriguing data, a couple of questions could be raised.

We thank the reviewer for the overall positive evaluation of our study.

1. From extensive network analysis, Btk was chosen as a new target. However, it is not known or addressed where Btk have impacts, while some analysis data suggesting the role of Btk in metabolic inflammation or inflammasome activation have been presented. It is recommended to conduct some *in vitro* experiment (even if preliminary) to explore the immunological steps affected by Btk.

We acknowledge the importance of investigating the biological processes in which BTK is involved to better understand its possible involvement in obesity. However, a literature search focused on this kinase pointed out a quite complex scenario. Indeed, along with B cells, *BTK* is expressed by different immune cells of the myeloid lineage relevant for obesity, such as macrophages and granulocytes, as shown in Supplementary Figure 7. At the molecular level, in addition to the role of BTK in B cell receptor signaling, BTK has been shown to be a component of the NLRP3 inflammasome (Ito et al. 2015; Liu et al. 2017). Moreover, additional literature showed the involvement of BTK in macrophage Toll-like receptor (TLR) signaling that leads to TNF production (Page et al. 2018 PMID: 29567473; Liljeroos et al. 2007 PMID: 17020802; Gray et al. 2006 PMID: 16439361). Given the complexity of the inflammatory response in obesity, with different cells and pathways involved, at this stage we think that is difficult to choose what to investigate *in vitro* (e.g. type of cell, protein expression). We therefore decided to start looking at the correlation between *Btk* expression levels and the infiltration of immune cells in *in vivo* models of metabolic inflammation. We chose zebrafish larvae as experimental organism to set up two high fat diet-induced obesity models and investigated *Btk* expression and macrophage accumulation. This model organism was chosen because it has the advantage of being easy to manipulate and analyze and at the same time has a fully developed innate inflammatory response. Although we are aware of the limitations of this model in reproducing the complexity of metabolic syndrome, we believe that it can give interesting hints about the potential use of BTK inhibitors for the treatment of diet-induced inflammation.

2. Ibrutinib was selected as a candidate for new drug repositioning. It is recommended to conduct experiments studying the effect of ibrutinib in metabolic inflammation/inflammasome activation *in vitro* and metabolic syndrome *in vivo*.

We followed the reviewer suggestion to strengthen the results of our computational work with a preliminary experimental validation. We have set up zebrafish models of diet-induced metabolic inflammation that were used to assess the potential of ibrutinib in reducing the high fat-induced intestinal inflammation. To this end, the accumulation level of macrophages of zebrafish larvae exposed to high fat diet was analyzed with/without adding ibrutinib to the embryo water.

3. Some more discussion regarding other seven drugs in Table 1 besides ibrutinib will be helpful.

Our focus on ibrutinib is a consequence of the filtering and prioritization procedure that identified this drug as the most promising candidate. In the adipose network, BTK, the target of ibrutinib, was the only target with a tissue-specific expression concordant with disease manifestation (obesity-related inflammation). We therefore focused on this result in the subsequent analyses. To clarify this, we expanded the results section and discussed the filtering and prioritization steps in more detail. The tissue expression of the other targets in the adipose network are reported in Supplementary Table 2. Moreover, we extended the analysis to investigate the results related to the liver and muscle networks, which have received little attention in the previous version of the manuscript. The tissue expression of the targets identified from liver and muscle networks are reported in Supplementary Table 3 and 4.

Reviewer #2 (Remarks to the Author):

In this manuscript, the authors describe a computational approach to finding new therapeutics for the metabolic syndrome through drug repurposing. Their proposed method is based on network analysis through integrating drug- and disease-related genes via various networks. The work led to a novel finding that ibrutinib is a potential new drug for the metabolic syndrome. The work certainly adds to the growing body of literature on computational drug repurposing through network-based analysis. The work is solid overall but could be further strengthened with regard to its main contributions to the technical advancement as this is mainly a methodology article in its current form. In addition, more lab work is certainly warranted for the further validation of their finding in practice. My specific comments can be found below:

1. The study provides some experimental evidence (in Fig 7) but as the authors concluded themselves “more experiments are needed to investigate ... the possible benefits for patients with metabolic syndrome.” Thus I believe the major contribution in this work lies in their proposed computational method. From the technical perspective, what is the unique novelty in your method vs. the previous network-based approaches. Highlighting such differences would be helpful for those interested in the algorithmic development and improvement. The generalizability of the method should also be extensively discussed.

We thank the reviewer for these comments, which were an impulse to contextualize our method and thereby clarify its novelty. To address the lack of comparison with previous network-based methods, we included an additional Supplementary Note that describes our method in the context of previous research in the field. We believe that the reader without specific knowledge in the research area can benefit from this section that refers to the most recent literature. Moreover, this note highlights the distinct features of our method facilitating the critical evaluation of the approach. On the other hand, the generalizability of our method has been included in the manuscript discussion. We provided examples of alternative applications of the pipeline by pointing out how the input data could be obtained from different sources.

The updated version of our work also includes the results of *in vivo* experiments that evaluated the repurposing potential of ibrutinib for diet-induced obesity. Our findings suggests a positive impact of ibrutinib in lowering the number of macrophages as described in the new paragraph added to the results section of the manuscript. See also our answer to reviewer 1, comment 1 and 2.

2. Because of the multi-step approach, I feel it is important for the authors to have evaluations of the derived data in each individual step. In other words, it is difficult to judge the data quality and completeness from the current description.

Thank you for this suggestion. The revised version of the manuscript includes two additional supplementary figures (Supplementary Fig. 3 and 4), that, together with Supplementary Fig. 2, allow to

easily follow the individual steps of the workflow. Moreover, we extended the results section of the manuscript to allow the reader to better distinguish each step.

3. It makes sense to supplement GWAS with literature mining. However, I'd say that using simple search with MeSH terms is not ideal (e.g. new articles are not indexed with MeSH). Furthermore, obtaining gene information from the list of MeSH substance is problematic as many gene/protein concepts are absent from MeSH (this is a known problem for MeSH vocabulary). Therefore, I'd suggest the authors either manually tag the genes from their search results or find an automated gene tagger.

As can be seen in Figure 1, there exist a number of genes, unique to text mining, even after filtering in the enrichment analysis. Have they been somewhat looked at at all?

We thank the reviewer for this comment that motivated us to revise this fundamental aspect of our pipeline and consequently update our strategy.

Regarding the search with MeSH terms, we agree with the reviewer that limiting the search to MeSH indexed articles can restrict the search space considerably. On the other hand, we believe that the identification of articles mainly focused on the disease of interest is crucial to successfully reconstruct the underlying biological processes. To this end, MeSH indexing can contribute to restrict the article search by using the PubMed search field tag [Majr:NoExp]. To take advantage of both MeSH and keyword search, we updated our search strategy as detailed in Materials and Methods. Briefly, the MeSH search was complemented with a keyword search using the PubMed [TIAB] tag to cover the recent literature not yet included in the MeSH indexing.

Regarding the gene annotation, we agree with the reviewer that the use of MeSH substances annotation to obtain disease-relevant genes was not optimal. Therefore, we followed the suggestion and used the automatic gene tagging provided by PubTator.

For the text-mining genes, we investigated their contribution to the final prediction as detailed in the Supplementary Note 2.

4. The authors stated the critical importance of combining data from different sources in various steps of their approach. e.g. combining GWAS with literature or merging PPI with transcriptional regulatory network. It would be helpful to tease out the effect of individual component for the final system prediction.

The revised manuscript includes a new Supplementary Note that highlights the contribution of the individual system components to the final prediction (GWAS vs. text mining, PPI vs. regulatory network). In addition to the suggested comparisons, we evaluated the effect of network-based distance vs. GO functional similarity to the overall prediction.

5. A known challenge in computational drug repurposing is selecting promising candidate drugs from the list of predictions made by an algorithm. Was your predication list manually examined and is this why only the results for the adipose network are investigated? Furthermore, how was the set of 8 drugs in table 1 identified and why the authors decided to further pursue Ibrutinib from the list (due

to experimental data availability?) Not sure if it is applicable but other types of algorithm validation that are worth looking into might be the use of patient EHR data (e.g. Cheng et al., Nat Comm, 2018).

We agree with the reviewer that in the first version of our manuscript the strategy to identify the most promising candidate drugs was not sufficiently well described. Moreover, by focusing only on the results derived from the adipose network, we might have lost interesting findings.

For these reasons, we have now added more details about the filtering and prioritization of the results in Materials and Methods as well as in the Results section. And the analysis was carried out for all three networks.

Briefly, starting from the drugs with a significant proximity score, we first excluded the drugs with an undesired side effect (related to MetSyn). Then, by focusing on the targets, we excluded drugs with a target already under investigation for MetSyn-related indications. Table 1 includes the drugs passing these two filtering steps. The only manual component of these two filtering steps has been the selection of the MetSyn related indications and side effects. Finally, we evaluated if the tissue expression of the targets (of the drugs in Table 1) was concordant with the disease manifestations. This prioritization step identified BTK, the target of ibrutinib, as the only protein in the adipose network with a tissue-specific expression relevant for the disease manifestation (obesity-related inflammation). The extension of the analysis to the other two networks pointed out erlotinib, targeting NR1I2, as the second promising repurposing candidate. However, a literature search related to this target revealed that the activation of the protein could contribute to the development of MetSyn and diabetes and this discouraged us from further investigating the repurposing potential of the drug.

With respect to the inclusion of clinical data in the prediction, we agree with the reviewer that the electronic health records represent an important source of information that can be exploited for drug repurposing. This is particularly true for widely used drugs, because the amount of available data is amenable for statistical analysis. However, an important limitation related to the use of EHR is the difficulty of effectively extracting the relevant information from mostly unstructured text. The text mining techniques needed to process these documents are still under heavy development and their application requires a lot of efforts. For this study we decided to focus on the integration of different type of clinical data, such as those derived from clinical trials and the FDA contraindications reported in Drugcentral db. However, we do not exclude that future extensions of the workflow could include EHR data.

6. According to the article, only 3 out of 6 MetSyn drugs were found to be significant in their analysis, suggesting the sensitivity of their approach is 50%. what about the other three? can you please discuss this in the paper.

We extended the results section of the manuscript as well as the Supplementary Note 3 to include a more detailed description of the score effectiveness.

Minor:

- Having a brief discussion of the results from the other two networks would be nice.

We agree that the inclusion of the discussion of the results obtained from the other networks improves the manuscript. The revised version includes a discussion for all three networks.

- I noted in a few cases where data were downloaded in 2016 (e.g. GWAS, text mining). Since it's fall 2018 now, would it make sense to update and use the latest version for your analysis?

We agree that in principle this could have affected the up-to-datedness of our work. We rerun the analysis with the most updated versions of the data (November 2018) and updated the manuscript accordingly. The main findings are not changed.

- Please replace comma with dot for the decimal scores in Table 1.

We modified the decimal scores and thank the reviewer for pointing this out.

Reviewer #3 (Remarks to the Author):

The authors proposed a drug repurposing approach based on network construction and analysis to identify therapies for metabolic syndroms.

The paper is missing of a proper introduction on the background of the data used in the approach. For example, why did the authors used GWAS? The authors should claim what are the pros and the cons of the data used. I think the use of the GWAS catalog and the SNPs related to the Metabolic Syndromes to start the network construction is an important and innovative step. The authors should discuss more about it since this is the starting point to construct the network.

We thank the reviewer for highlighting the importance of using GWAS derived data to characterize the disease of interest and identify repurposing candidates. In the Discussion section we refer to literature supporting this choice. Furthermore, we added a paragraph in the Results section to point out the pros and cons of GWAS data for the disease characterization and drug effects analyzes. However, we would like to clarify that the networks are constructed using transcriptional regulatory and protein-protein interaction networks, while the GWAS results are used to identify disease modules in the networks.

The paper lacks comparisons about different approaches both toward the drug repurposing and network analysis metric (proximity score).

Examples:

- Cheng, Feixiong, et al Nature communications 9.1 (2018): 2691

- Vitali, F., Journal of biomedical informatics, 46(5), 876-881.

The novelty of the paper it is unclear.

We agree with the reviewer that the previous version of our manuscript was lacking a comparison with other approaches. To address this and clarify the novelty of our method, the revised manuscript now includes an additional Supplementary Note that describes our method in the context of previous network-based methods. See also our answer to Reviewer 2, comment 1.

The results are difficult to follow without reading the methods as a lot of details on the adopted procedures are missing.

We updated the Results section to enhance the readability of the manuscript. For example, we now describe the source of the data as well as the filtering and prioritization of the repurposing candidates in more detail. We hope that this will help the reader to follow the Results section easily.

- The authors identified the starting genes from the GWAS Catalog, another study (Supplementary Table 1) and text mining. In Fig 2a is shown that most of the genes came from the text-mining search. This type of evidence can be considered less reliable if compared to the other information. How the use of this type of evidence affect the results?

We agree with the reviewer that theoretically results coming from the text-mining search could be considered more uncertain. However, we would like to highlight that in our workflow the final selection of text mining derived genes is guided by the GWAS results through pathway analysis. We decided to include text mining data to cope with the incompleteness of the GWAS derived data. Indeed, our analysis showed that the addition of text mining genes is improving the characterization of the disease. To address the contribution of the different data sources (GWAS vs. text-mining) to the final prediction we included a new Supplementary Note (Supplementary Note 2).

- How the tissues were included in the network? Where this information came from? The authors should make it clearer in the results as well

We updated the Results section to include this information. The tissue-specific regulatory networks were directly obtained from regulatory circuits, which are inferred using FANTOM5 data. The tissue-specific PPI networks were created using HIPPIE db, by restricting to proteins expressed in the relevant tissue based on GTEx data.

- Line 98: how the enrichment has been performed? Using Reactome?

We thank the reviewer for pointing this out. We updated the manuscript with a more detailed description of the databases used. The enrichment analysis was carried out using enrichR, while the following databases have been considered: GO biological processes, KEGG, WikiPathways, Reactome, Biocarta, Humancyc, NCI-Nature, and Panther.

- Line 147 – how the information on target have been integrated?

We agree with the reviewer that in the first version of our manuscript the strategy to identify the most promising candidate drugs was not sufficiently well explained. We updated the Results and Materials and Methods sections to provide a more detailed description of the filtering and prioritization steps. The target information from Open Targets platform was used to identify (and exclude) those targets already under investigation for therapeutic interventions related to metabolic syndrome. Please also refer to our answer to reviewer 2, comment 5.

- Figure caption and number are missing.

Figure captions and legends can be found after the references and competing interest statement as indicated in the Nature Communication Submission guidelines. We hope that the reviewer did not receive a disrupted document.

The results should be more robust, for example the authors claimed (line 145) that in the adipose network 3 out of 6 MetSyn drugs were identified thanks to their approach. The authors should state how they identified the six drugs and which are these drugs.

We extended the main text as well as Supplementary Note 3 to include a more detailed description of the score effectiveness.

The authors proposed to use a proximity score that combines network distance with semantic similarities, they should justify better their choice since other methods already exist for that aim (e.g. Vella, D (2018). Scientific reports, 8(1), 5499)

We agree with the reviewer that a comparison of our method with previous ones is important. Therefore, we added a comparison with previous methods that describe our method in the context of previous research in the field (Supplementary Note 1). Regarding the publication by Vella et al., we would like to point out that the proposed method is pursuing another aim compared to our study. While the MTGO algorithm aims at the detection of functional modules in PPI networks using topological information and GO knowledge, the proximity score we defined is used to identify connections between modules.

The authors validate one drug candidate out of the 8 drugs identified as potential by analyzing the expression of its target in public gene expression data of obese patients.

This limits the impact of the paper. Additional comments on the other drugs identified should be provided. Also, there is no experiment in verifying the candidates.

It is correct that we did not discuss all drugs listed in Table 1. However, this was due to the fact that our filtering and prioritization procedure identified ibrutinib as the most promising candidate and therefore we focused on this result. To clarify this, we expanded the Results section and discussed the filtering and prioritization steps in more detail. See also our answer to reviewer 1, comment 3.

We agree with the reviewer that the inclusion of experiments to validate the systems biology findings would have a positive impact on our study. For this reason we performed *in vivo* experiments using zebrafish larvae to assess the effect of ibrutinib on a model of diet-induced obesity. We show that ibrutinib lowers the macrophage infiltration of zebrafish larvae. A new paragraph has been added to the results section to describe the new findings. Despite the intrinsic limitations of this model, we believe that it provides some interesting hints that can be further explored by other studies. See also our answer to reviewer 1, comments 1 and 2.

Other minor comments:

- Line 73: GWAS reference missing

We added the reference and thank the reviewer for pointing this out.

- The authors should provide the figures on the three-tissue network identified, and preferably their node and edges as supplementary files.

Unfortunately, the networks are too big to provide a meaningful figure of the complete networks. However, we agree that providing the list of edges/nodes improves the reproducibility of our work and we therefore added an additional supplementary data file (Supplementary Data 4).

- In Computational framework overview: the authors should reference their approach steps with the number reported in Fig1.

Thank you for the suggestion. We agree that this will improve the readability and included the respective numbers.

Reviewers' comments:

Reviewer #1 (Remarks to the Author):

This revised paper entitled "Metabolic syndrome: identification of deregulated pathways and drug effects by network analysis" by Misselback K et al. added some more data according to the reviewers' suggestions. While this paper presents a novel approach and added some more data during revision, still it is hard to understand the conclusion.

1. The mechanism underlying the effect of Btk on inflammasome activation has not been addressed. The results in Fig. 7C (decreased Btk expression in adipose tissue of caspase 1-KO but unaltered Btk expression in ASC1-KO) are hard to understand. The reason why Btk expression is increased in adipose tissue of HFD-fed GPR120 KO is not clear and not addressed.
2. The in vivo role of ibrutinib was conducted using a zebrafish model. However, only the number of macrophages in larvae was studied. In zebrafish larvae or adult stage, further analysis for additional metabolic parameters such as body weight, fasting blood glucose, blood cholesterol, blood TG, adipose tissue size or H&E staining/gene expression in adipose tissue) has been performed by other researchers. If the authors want to use zebra fish model as a tool to test the metabolic efficacy, it is recommended to investigate beneficial effects of BTK inhibitor on metabolic profile of zebra fish.
3. In the Reference section, is reference 38 related to GFR120 KO?

Reviewer #2 (Remarks to the Author):

I thank the authors for addressing the previous reviewers' comments, through more detailed clarification as well as additional analyses and experiments. Overall, I find their responses to be proper and satisfactory. As a result, the manuscript has improved accordingly.

I particularly find it interesting in the new analysis results regarding the impact of different method components. The supplementary tables 1-4 clearly show the contributions of individual data sources in the results of each intermediate step. But this also left me wondering whether the final computational results would be significantly different (e.g. ibrutinib would not be identified anymore) if one of the contributing data source in genes identification (gwas catalog; gwas summary statistics, text mining) or tissue-specific network generation (regulatory networks vs. PPI networks) was taken out from the study design. For example, if the inflammation-related MetSyn module in the adipose network was absent due to missing genes from literature, would you still be able to identify ibrutinib? Briefly shedding some light on this would be important and help highlight your novel design in your method.

Other minor points:

- It would be helpful to establish crosslinks between figures in the main article to those details in the supplementary data, which provide more complete picture of the analysis.
- ,diabetes' search-term in supplementary data; fix punctuation
- The size of venn diagrams is not seemingly proportional to the actual numbers inside (e.g. figure 2a). please adjust their size accordingly.
- Line 380: reference of the new TM tool pubtator does not seem correct.
- Line 454: "Wang method" should be cited when it first appears.

Reviewer #3 (Remarks to the Author):

The authors well addressed the reviewer answers. In particular, they performed additional experiments to validate the drug candidate identified by their method.

In the discussion, the authors should include the limitation of their methodology, potential extensions and future prospectives.

Before publication, the paper should be revised to avoid repetitions between the Introduction Results and discussion section.

We would like to thank the reviewers for their comments that encouraged us to further improve the manuscript. Point-by-point responses are reported below.

Reviewers' comments:

Reviewer #1 (Remarks to the Author):

This revised paper entitled “Metabolic syndrome: identification of deregulated pathways and drug effects by network analysis” by Misselback K et al. added some more data according to the reviewers' suggestions. While this paper presents a novel approach and added some more data during revision, still it is hard to understand the conclusion.

1. The mechanism underlying the effect of Btk on inflammasome activation has not been addressed. The results in Fig. 7C (decreased Btk expression in adipose tissue of caspase 1-KO but unaltered Btk expression in ASC1-KO) are hard to understand. The reason why Btk expression is increased in adipose tissue of HFD-fed GPR120 KO is not clear and not addressed.

We thank the reviewer for the pointing us to a need of clarity regarding Btk expression in adipose tissue. We addressed this point by adding a better description of the gene expression datasets and by clearly stating the conclusion of this analysis. The new version of the manuscript includes findings derived from the publications in which these datasets were originally described that we think are helpful to understand the modulation of *Btk* expression in the context of obesity and metabolic dysregulation. Briefly, we chose these datasets to investigate the association between *Btk* expression and macrophage infiltration in the adipose tissue and we could conclude that the increase of *Btk* mRNA in adipose tissue is dependent on the presence macrophage infiltration.

2. The in vivo role of ibrutinib was conducted using a zebrafish model. However, only the number of macrophages in larvae was studied. In zebrafish larvae or adult stage, further analysis for additional metabolic parameters such as body weight, fasting blood glucose, blood cholesterol, blood TG, adipose tissue size or H&E staining/gene expression in adipose tissue) has been performed by other researchers. If the authors want to use zebra fish model as a tool to test the metabolic efficacy, it is recommended to investigate beneficial effects of BTK inhibitor on metabolic profile of zebra fish.

We thank the reviewer for the useful suggestions to address the changes in lipid metabolism observed in the zebrafish models, which has helped us to add additional evidence for the validity of our model. Due to the impracticalities in measuring blood metabolites levels in larvae, we have now extended our study to include an evaluation of fat deposition by using two different and well established staining methods for lipid accumulation in zebrafish (Nile Red and Oil Red O staining), as well as a qPCR assessment of a key regulator of lipid metabolism driving de novo fatty acids and triacylglycerol synthesis (ie *srebf1*) and the inflammatory marker *il1b*. Both staining approaches confirmed a significant increase of lipid accumulation in comparison with the standard diet upon High Fat and High cholesterol diet. Stereomicroscopic analysis of Oil Red O stained larvae allowed to classify them according to three different levels of fat accumulation, while with confocal microscope analysis of Nile Red stained larvae revealed the different accumulation of fat droplets in the subintestinal space, probably in blood vessels,

with HCD, and in the enterocytes with HFD. We also found an increase in expression of key genes related to lipid metabolism (*srebf1*) and inflammatory responses (*il1b*) as measured by qPCR analysis. Beneficial effects of ibrutinib treatments were revealed in all our analyses, with the exception of fat accumulation, where only a slight, but not significant, reduction in high lipid accumulation larvae was observed by Oil Red O staining, while *srebf1* expression as well as inflammation markers were returned to normal value by ibrutinib treatment. Our interpretation is that the btk inhibitor acts quickly by blocking the inflammatory response, whereas the effects on fat metabolism, that are documented by the reduction of *srebf1* expression, requires longer to be appreciated.

3. In the Reference section, is reference 38 related to GFR120 KO?

We thank the reviewer to point us to this mistake, that we have now corrected in the revised version. To help the reader in finding the source datasets, we have also added their NCBI GEO accession codes.

Reviewer #2 (Remarks to the Author):

I thank the authors for addressing the previous reviewers' comments, through more detailed clarification as well as additional analyses and experiments. Overall, I find their responses to be proper and satisfactory. As a result, the manuscript has improved accordingly.

I particularly find it interesting in the new analysis results regarding the impact of different method components. The supplementary tables 1-4 clearly show the contributions of individual data sources in the results of each intermediate step. But this also left me wondering whether the final computational results would be significantly different (e.g. ibrutinib would not be identified anymore) if one of the contributing data source in genes identification (gwas catalog; gwas summary statistics, text mining) or tissue-specific network generation (regulatory networks vs. PPI networks) was taken out from the study design. For example, if the inflammation-related MetSyn module in the adipose network was absent due to missing genes from literature, would you still be able to identify ibrutinib? Briefly shedding some light on this would be important and help highlight your novel design in your method.

We thank the reviewer for the overall positive feedback. To address the final comment, we added a paragraph to the Discussion in which we underline that without text-mining derived genes we would not have identified ibrutinib as a candidate drug for repurposing.

Other minor points:

- It would be helpful to establish crosslinks between figures in the main article to those details in the supplementary data, which provide more complete picture of the analysis.

Thank you for the suggestion. When relevant, at the end of the figure captions we added a link to the Supplementary data.

- ,diabetes' search-term in supplementary data; fix punctuation

Thank you for pointing this out. We have now corrected this typo.

- The size of venn diagrams is not seemingly proportional to the actual numbers inside (e.g. figure 2a). please adjust their size accordingly.

We modified Figure 2a and 3b according to the comment.

- Line 380: reference of the new TM tool pubtator does not seem correct.

We corrected it.

- Line 454: "Wang method" should be cited when it first appears.

We have now fixed the citation.

Reviewer #3 (Remarks to the Author):

The authors well addressed the reviewer answers. In particular, they performed additional experiments to validate the drug candidate identified by their method.

In the discussion, the authors should include the limitation of their methodology, potential extensions and future prospectives.

We followed the reviewer suggestion. A new paragraph has been added to the Discussion section.

Before publication, the paper should be revised to avoid repetitions between the Introduction Results and discussion section.

We checked the manuscript for repetitions and removed them.

REVIEWERS' COMMENTS:

Reviewer #1 (Remarks to the Author):

Since this manuscript has been improved by the incorporation of the reviewers' comments, now this paper is acceptable for publication.

Reviewer #2 (Remarks to the Author):

My previously raised points have been adequately addressed in this revised manuscript. I have no further comments.

Reviewer #1 (Remarks to the Author):

Since this manuscript has been improved by the incorporation of the reviewers' comments, now this paper is acceptable for publication.

Reviewer #2 (Remarks to the Author):

My previously raised points have been adequately addressed in this revised manuscript. I have no further comments.

We thank again the reviewers for their comments and suggestions that helped us to improve the manuscript.